# PROGRAMMATIC REPRESENTATION LEARNING WITH LANGUAGE MODELS

## ABSTRACT

Classical models for supervised machine learning, such as decision trees, are efficient and interpretable predictors, but their quality is highly dependent on the particular choice of input features. Although neural networks can learn useful representations directly from raw data (e.g., images or text), this comes at the expense of interpretability and the need for specialized hardware to run them efficiently. In this paper, we explore a hypothesis class we call *Learned Programmatic Representations* (LeaPR) models, which stack arbitrary features represented as code (functions from data points to scalars) and decision tree predictors. We synthesize feature functions using Large Language Models (LLMs), which have rich prior knowledge in a wide range of domains and a remarkable ability to write code using existing domain-specific libraries. We propose two algorithms to learn LeaPR models from supervised data. First, we design an adaptation of FunSearch to learn *features* rather than directly generate predictors. Then, we develop a novel variant of the classical ID3 algorithm for decision tree learning, where new features are generated on demand when splitting leaf nodes. In experiments from chess position evaluation to image and text classification, our methods learn high-quality, neural network-free predictors often competitive with neural networks. Our work suggests a flexible paradigm for learning interpretable representations end-to-end where features and predictions can be readily inspected and understood.

## 1 INTRODUCTION

The central problem in supervised machine learning is to find a predictor $h : X \to Y$ in a hypothesis class $\mathcal{H}$ that minimizes a certain risk function $\mathcal{R}(h)$, such as $0 - 1$ error in classification or mean-squared error in regression (Michalski et al., 2013). Classical choices for $\mathcal{H}$ include linear models, decision trees, and ensembles thereof, which are compellingly simple to understand and debug, and are both compute- and data-efficient. However, their effectiveness is highly limited in domains with unstructured, high-dimensional inputs, such as images or text. For these domains, high-quality models are often best learned by first constructing a high level *representation* of an input $x \in X$ using a set of features $f_i : X \to \mathcal{R}$ that yield a higher-level encoding of the input that predictors can then rely on. While this offers great flexibility, in practice the effort and domain expertise required to *engineer* a good set of features for a particular learning task severely limits the quality of models that can be obtained with classical predictors in high-dimensional input domains without extensive human effort (Dong & Liu, 2018; Cheng & Camargo, 2023).

A remarkably successful paradigm that avoids the need for hand-designed feature engineering is *deep learning*, where $\mathcal{H}$ is set to a parameterized family of neural networks of a domain-appropriate architecture. The core advantage of deep learning is the ability of gradient-based optimization to automatically learn useful representations from raw data (Bengio & LeCun, 2007; Damian et al., 2022). Indeed, deep neural networks can be seen as computing a set of complex, non-linear neural features, then applying a simple predictor on top (e.g., the last fully-connected layer, corresponding to a linear model). However, despite being highly effective for *prediction*, neural features have several drawbacks. First, deep neural networks trained as end-to-end predictors are data-intensive, and their ability to generalize drops drastically when in-domain data are scarce (Wang et al., 2023). Second, neural features are not easily interpretable: analyses of large-scale neural networks typically only find a fraction of neurons that seem to encode human-aligned concepts (Huben et al., 2023). This limits the potential of neural models to provide faithful *explanations* for their predictions (e.g., express *why* a given $x$ is being classified as $y$), which are important when experts rely on learned models to support high-stakes decision-making (Doshi-Velez & Kim, 2017).

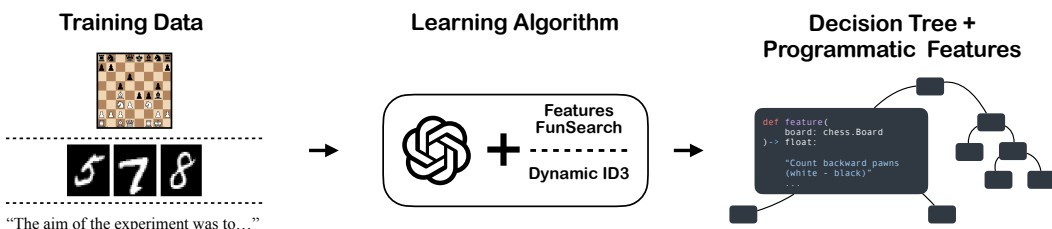

Figure 1: Learned Programmatic Representation models combine *programmatic features*, synthesized by LLMs as code, and decision tree predictors, yielding interpretable models. We give two algorithms for learning them end-to-end from supervised data in high-dimensional input domains.

In this paper, we seek to investigate alternative paradigms for representation learning that, like deep learning, do not require manual feature engineering from users and, like classical methods, yield interpretable, fast, and efficiently-learnable predictors. To that end, we propose learning *LeaPR* (Learned Programmatic Representation) models: a class of predictors with programmatic features paired with decision tree predictors, as illustrated in Figure 1. We leverage Large Language Models' (LLMs) ability to generate code using domain-specific libraries to encode features as arbitrary Python functions from the input domain into the reals: these functions are generated during training with the goal of improving the empirical risk of the current predictor.

For learning from supervised data, we propose two alternative methods. First, we explore a variant of the FunSearch method (Romera-Paredes et al., 2024) called F2 (for **F**eatures **F**unSearch). In F2, an LLM iteratively generates functions that compute features from the input domain, which are then evaluated by training a Random Forest (Breiman, 2001) predictor and extracting importance weights. These scores are used in-context to steer the LLM to generate features with high importance that are thus as predictive as possible. F2 is a *black-box* procedure with respect to the underlying classic predictor, leveraging an off-the-shelf Random Forest training method. We further introduce a *white-box* training procedure for LeaPR models called Dynamic ID3, or D-ID3, inspired by the classical ID3 algorithm for training decision trees. D-ID3 follows an ID3-like training loop where leaf nodes of a growing decision tree are selected and "split" by introducing two new leaves and a decision based on the value of a specific feature. While ID3 operates with pre-existing features, D-ID3 queries an LLM to propose new programmatic features on the fly that can help split the node under consideration. D-ID3 gives the LLM the decision path leading to the node, with few-shot examples of samples in that branch in domains where inputs can be encoded as text, and asks for novel features that are useful for that specific context. The vast prior knowledge of modern LLMs about domain-specific Python libraries makes both approaches highly applicable across diverse, high-dimensional input domains.

We evaluate both methods on two LLMs across three domains: chess position evaluation, image classification (MNIST (LeCun, 1998) and Fashion-MNIST (Xiao et al., 2017)), and text classification (AI vs. human text classification on the Ghostbuster dataset (Verma et al., 2024)). Across domains, LeaPR methods learn high-quality neural network-free representations that extract empirically useful features, often spanning tens of thousands of lines of Python code spread across hundreds of functions, all stitched together by decision trees. The learned features tend to be highly intuitive yet specific enough to lead to high-quality predictors. Our main contributions are:

- We propose jointly learning *programmatic features*, represented as LLM-generated functions from the input domain to the reals, together with decision tree predictors, thus obtaining fast, interpretable predictors.

- We introduce F2 and D-ID3, two algorithms for learning LeaPR models, where features are generated on demand during decision tree training.

- We evaluate LeaPR models on three domains: chess positions, images and text, showing comparable accuracy and often favorable data efficiency compared to baseline methods. We analyze the learned features and show how programmatic features can be useful for data exploration and for understanding model failures.

## 2 RELATED WORK

**Code generation with LLMs** Our work is enabled by the capability of LLMs to generate code under flexible task specifications, including natural language instructions and examples (Chen et al., 2021). This ability has been explored in a variety of domains, including using code as an intermediate tool for reasoning Chen et al. (2022) and as an interface for agents to interact with an external environment Lv et al. (2024). We exploit code generation for synthesizing *features* from arbitrary input domains, building on LLMs' prior knowledge of a rich set of existing libraries. Feature generation, both with and without LLMs, has been widely explored in tabular datasets (Ko et al., 2025; Zhang & Liu, 2024), whereas we explore domains with more complex, unstructured input (such as images and text), where current methods do not apply. We provide an extensive discussion of existing feature generation methods in Appendix D.

**Black-box code evolution with LLMs** LLMs can also be used to *optimize* a black-box objective function (Lange et al., 2024). This approach was pioneered in FunSearch (Romera-Paredes et al., 2024), the method that inspires our F2 method (Section 3.1). In FunSearch, the LLM is given the type signature of a function to synthesize, and it outputs candidate functions which are scored with a metric unknown to the model. LLMs can then see past candidate function examples and their scores in following rounds (Romera-Paredes et al., 2024), and can mutate and evolve previous attempts to propose new, more successful ones. AlphaEvolve (Novikov et al., 2025) explored this idea with larger models and novel methods to ensure diverse candidates. Our work differs from FunSearch and AlphaEvolve in that LeaPR models use LLM-generated code simply to generate *features*, and not the full predictor end-to-end. Using classical models to utilize LLM-generated modules allows our method to scale and simultaneously employ thousands of features at once: for instance, yielding predictors at much a larger scale magnitude than what AlphaEvolve has been used to generate (e.g., up to 50k lines in total in the chess features used in our largest predictor).

**Learning programmatic world models** A modular approach for using LLM-synthesized code has been recently employed in PoE-World (Piriyakulkij et al., 2025) for the problem of learning *world models*. In PoE-World, LLMs generate a set of small predictors that individually explain a specific behavior of the environment; a separate model combines these predictors with weights learned by gradient descent. This scales better than monolithic code representations, which World-Coder introduced earlier (Tang et al., 2024). Like LeaPR models, PoE-World can generate world models with thousands of lines since LLMs do not have to generate complete programs, only small modules that can be composed scalably. We propose a similarly modular architecture (scaling to tens of thousands of lines) aimed toward the general problem of supervised learning. The observation that synthesis of large programs scales better by learning libraries has a long history in program synthesis, including in DreamCoder (Ellis et al. (2021), with libraries constructed via symbolic compression) and Lilo (Grand et al. (2023), with LLM-generated abstractions).

**Interpretability of neural networks** The widespread deployment of deep neural networks strongly motivated the research community to understand the behavior of neural models (Mu & Andreas, 2021), often through the lens of *mechanistic interpretability* Conmy et al. (2023). One key hypothesis is that neural networks have interpretable "circuits" that drive specific behaviors (Marks et al., 2025) or encode learned knowledge (Yao et al., 2025). Prior work has also explored neural architectures that are more amenable to interpretation by construction (Doumbouya et al., 2025; Hewitt et al., 2023), rather than post-hoc. Several works also explore reshaping the prediction pipeline to lead to more interpretable decisions, including ViperGPT (Surís et al., 2023), Tree Prompting (Singh et al., 2023), and FM+V-IP (Chan et al., 2023). In all of these, however, neural networks are still involved at inference time, and even in cases involving code generation, such as in ViperGPT, the code representes a full predictor, and not a feature as in LeaPR.

## 3 LEARNING PROGRAMMATIC REPRESENTATIONS

Classical predictors considered in supervised machine learning, like decision trees, are *intrinsically explainable* in the sense that their structure is simple enough to warrant human inspection: this is highly desirable when humans might want to trust (e.g. in high-stakes decision making) or learn from (e.g., someone learning to play chess) machine learning models. However, this simplicity

comes at a steep cost: the performance of these methods is notoriously impacted by the particular choice of features given as their input.

Consider learning a board evaluation decision tree for the game of chess: a predictor for the likelihood that the player with the white pieces will win. During inference, a decision tree tests the value of one input variable at a time. If given only the raw information available on the board (e.g. the piece lying on each of the 8x8 squares), each input variable alone carries negligible information about the overall position. Leaf nodes are forced to be invariant to all features not tested on the path from the root to that node; thus, if any decision is made before testing *all* 64 squares, critical information can always be missed (e.g., a queen on one of the unobserved squares). Thus, these individual features are not informative enough for decision trees to encode effective predictors.

On the other hand, if we instead train the model on a *single useful feature* computed from the board, like the *material difference* between players (given by a weighted sum of the pieces of each color still on the board), even a very shallow decision tree can make predictions that are significantly better than random, since a large material advantage is highly correlated with one's chance of winning. Adding more informative features will progressively allow a decision tree learner to improve.

In this paper, our starting point is the insight that LLMs have two capabilities that allow them to serve as highly effective *feature engineers* for classical ML models. First, many useful features, such as the material balance in chess described above, can be implemented in a few lines of code, and modern LLMs are capable of flexibly generating code to accomplish a variety of tasks. Second, broad pre-training of LLMs encodes useful prior knowledge about a very wide range of domains (Bommasani et al., 2021), equipping them with strong priors over what kinds of features could be useful for making predictions across these domains. Our main hypothesis is that we might be able to train high-quality classical models by leveraging LLM-generated features at scale. Our goal here is thus to learn a hypothesis class that consists of (a) *programmatic features* and (b) decision tree predictors. We call these *Learned Programmatic Representation* models, or *LeaPR* models. The main challenge we now tackle is how to elicit predictive features from language models.

### 3.1 BLACK-BOX FEATURE GENERATION

Given a supervised dataset $\mathcal{D}$ and any scoring function $S_{\mathcal{D}}(f)$ that measures the quality of a proposed *feature* $f : X \to \mathbb{R}$, a simple methodology to obtain increasingly good features is to apply a FunSearch-style procedure, where an LLM is used to propose candidates to try to maximize $S$. Modern LLMs are capable of generating complex functions even for high-dimensional input domains, such as images and text, partly due to their ability to write code that uses existing human-written libraries for each of these domains. Thus, the main component needed for this approach to work is to answer: what makes $f$ a good feature?

In the standard FunSearch setup (Romera-Paredes et al., 2024), the scoring function $S$ evaluates candidates independently: proposals are *self-contained solutions* of the task. Our setup here is different: for the purpose of learning a predictor from *all* generated features at once, a new candidate $f_k$ is valuable only to the extent that it contributes predictive information when taking into account the existing feature set $f_{1:k-1}$. Assuming that we keep track of the set of features generated so far and propose new features in a loop, a naïve adaptation of FunSearch would thus score a new feature $f_k$ on its *added* predictive power once it has been proposed. For that, we could train a new predictor using $f_{1:k}$ and compare its risk against the previous predictor trained on $f_{1:k-1}$. However, this runs into the issue that early features receive disproportionately high scores simply because their baselines (initial predictors based on few features) are severely limited. In practice, with this approach, the highest-scoring features are essentially fixed after the first few iterations, which is undesirable: we would like to detect and reward powerful features even if they appear late during training.

To overcome this problem, we *simultaneously score all existing features* $f_{1:k}$ independently of the order in which they were proposed. Given a learned decision tree, prior work has proposed several metrics of *importance* of each input feature (e.g., measuring decrease in "impurity" in decision nodes that use a given feature, (Breiman, 2001)). Importance metrics only depend on the final learned predictor, and decision tree learning methods are order-invariant with respect to input features.

Algorithm 1 (Figure 2, left) shows *Features FunSearch* (F2), our representation learning algorithm based on this FunSearch-style approach but with features scored as a set. Specifically, F2 takes

**Algorithm 1:** Features FunSearch (F2)

**Input** : Language model, $LM$,
   Supervised dataset $D \in 2^{X \times Y}$
**Output:** List of features $F$, where $f_i : X \to \mathbb{R}$
$F \leftarrow [\,]$;
**for** $iteration \in [1, \cdots, T]$ **do**
 $rf \leftarrow$ TrainRandomForest$(D, F)$ ;
 $imp \leftarrow$ FeatureImportances$(rf)$ ;
 $top\_k \leftarrow$ TopKFeatures$(F, imp)$;
 $r \leftarrow$ RandomKFeatures$(F \setminus top\_k, imp)$;
 $p \leftarrow$ ProposeFeatures$(LM, top\_k, r)$;
 $F.\texttt{extend}(p)$;
**end**
**return** $F$ ;

**Algorithm 2:** Dynamic ID3 (D-ID3)

**Input** : Language model $LM$,
   Supervised dataset $D \in 2^{X \times Y}$
**Output:** List of features $F$, where $f_i : X \to \mathbb{R}$
$T \leftarrow \texttt{Leaf}(D_{train})$;
**for** $iteration \in [1, \cdots, T]$ **do**
 $l \leftarrow \arg\max_{\texttt{IsLeaf}(n)}$ TotalError$(T, n.data)$;
 $p \leftarrow$ ProposeFeatures$(LM, l.path\_to\_root)$ ;
 $\langle f, t \rangle \leftarrow$
  $\arg\min_{f \in F, t}$ SplitError$(f, t, n.data)$;
 $l.\text{split}(f, t, \{x \in n.data | f(x) < t\},$
     $\{x \in n.data | f(x) > t\})$ ;
**end**
**return** $\{n.\text{splitting\_feature} | n \in T \wedge \text{Internal}(n)\}$ ;

Figure 2: Two learning algorithms for LeaPR models. F2 (left) uses a FunSearch-style loop that attempts to evolve features that are *globally useful* to train a Random Forest predictor, as estimated by feature importances. D-ID3 (right) runs an ID3-style decision tree training loop and attempts to propose new features that are *locally useful* for splitting specific leaf nodes, attempting to minimize their impurity (e.g., variance in regression, or entropy in classification). Full subroutine definitions in Appendix A.

a supervised dataset and learns a programmatic representation — i.e. a set of feature functions $f_i : X \to \mathbb{R}$, represented as executable code. Like FunSearch, F2 iteratively uses an LLM to make batches of proposals conditioned on a sample of existing features, which are shown to the model along with their assigned scores — the LLM's task is to propose new features that will be assigned high importance score in a newly trained Random Forest predictor. These scores are a *global estimate* of the predictive power of each feature in a predictor trained with all of them.

## 3.2 Dynamic splitting

While F2 uses an underlying decision tree learner as a black-box, the insight that LLMs can be used to generate features on demand can serve as the basis for designing the decision tree learner itself. Recall that during inference in a decision tree, we start at the root and repeatedly follow the "decisions" associated with each node until we reach a leaf. Each such decision consists of testing the value of a particular feature: if this node splits on feature $f_k$, we compare $f_k(x)$ with a threshold $t$ learned during training. If $f_k(x) < t$, the node recursively returns the prediction made by its left child (or right child if $f_k(x) \geq t$). Leaf nodes return a fixed prediction defined during training, e.g., the most common class label (classification), or average value (regression) for training points that fall on that leaf. For training, classical decision tree learning algorithms (e.g., ID3 or CART (Quinlan, 1986)) start with a single node and repeatedly improve the current decision tree predictor by (a) choosing a leaf node and (b) partitioning it into a new decision node with two new leaves as its children. Partitioning searches for a feature and comparison threshold that minimize the "impurity" (e.g., variance in the continuous case, or entropy of class labels in classification) of data points falling on both sides. For instance, in classification, the best-case scenario would be to find a partition where all training data points falling on each new leaf belong to the same class.

However, the ability of classical algorithms to find good splits is limited by the predictive power of preexisting dataset features. Here, we revisit this recursive splitting strategy considering that we can attempt to generate new features *on demand* for the purpose of successfully partitioning a particular leaf node. When we decide to split a leaf, we have significant local context aside from global dataset information: in particular, we know the specific path of decisions that leads to that leaf, and we have a corresponding set of training examples, with their labels, falling onto that node. To be *locally useful*, a feature only needs to help distinguish between examples in that set. Indeed, for informing the proposals of potentially useful features, we can even leverage the ability of LLMs to performe inductive reasoning, by presenting actual examples (if possible in text), along with their labels, in the model's context: its objective, then, is to propose a feature that would explain the variation in the labels between those examples and others that reach the same leaf.

Algorithm 2, Dynamic ID3 (D-ID3), realizes this idea. In each iteration, D-ID3 selects the current leaf in the tree that accounts for the largest portion of training error (e.g. number of misclassified training examples). D-ID3 then generates new candidate features with an LLM on the fly to split that particular leaf. In modalities where we can easily represent examples in text, the LLM receives a sample of examples and their labels that fall in this branch (in our experiments, this only excludes image classification, where we only show a sample of image *class labels* in the prompt). D-ID3 considers these features, as well as all candidate features generated for ancestor nodes, and finds the best split for this leaf according to a user-defined impurity metric (all metrics available for classical methods are also possible here). This process repeats for a number of iterations. At the end, like F2, D-ID3 returns a learned *representation*: the set of programmatic features for the input domain that were used in the resulting decision tree. We note several practical considerations for both F2 and D-ID3, as well as other implementation details, in Appendix A.2.

## 4 Experiments

We now evaluate programmatic representation models on three tasks with complex input domains where the standard practice is to train neural networks: chess position evaluation (given a chess board, predict the probability that White wins), image classification on MNIST (LeCun, 1998) and Fashion-MNIST (Xiao et al., 2017), and text classification (detecting whether a piece of text is human- or AI-generated) on Ghostbuster (Verma et al., 2024). In all domains, we compare LeaPR against standard neural network baselines; additionally, in chess and image classification, we also include the Random Forest baseline where we feed a simple "raw" encoding of the input (the piece in each square for chess boards, or pixel values for images). We also compare against a baseline FunSearch method that learns the full predictor with code evolution, as well as a Program of Thoughts baselines – both of these using GPT-5.1 as their underlying language model (details in Appendix B.3). These baselines can also use the rich prior knowledge of language models and their code generation capabilities, but without the modularity that the LeaPR paradigm provides. We discuss the features our methods learn in each domain, and finally conduct a case study debugging a classifier that has learned to rely on a spurious feature in the Waterbird dataset (Sagawa et al., 2020).

We run F2 and D-ID3 using two OpenAI models, for a total of 4 LeaPR models per task: GPT 4o-mini `gpt-4o-mini-2024-07-21` (Hurst et al., 2024) and GPT 5-mini `gpt-5-mini-2025-08-07` (OpenAI, 2025). We run both methods so that they output a maximum of 1000 features — this means using 1000 iterations of D-ID3, and 100 iterations of F2 with a proposal batch size of 10 features in each call. We sometimes end with fewer than 1000 features because we discard features that fail validation (see Section A.2) Using the features learned by either algorithm, we then train a Random Forest model using the standard Scikit-Learn (Pedregosa et al., 2011) implementation, with 500 trees and a maximum depth of 50.

### 4.1 Chess position evaluation

First, we train models on the regression task of state-value prediction in the game of chess: given the board position, predict the win probability for each player. We use a publicly available dataset of games from the Lichess online platform (Lichess.org), and hold out 1000 random board positions for evaluation. The dataset comes with state values estimated by Stockfish (Romstad et al., 2008), the strongest publicly available chess engine. We use Stockfish's prediction value as the ground truth (Stockfish outputs values in "centipawns", which we convert to win percentages using the standard formula used by Lichess and other prior work). To represent and manipulate chess boards, we use the popular `python-chess` library, with a standard API that facilitates iterating through the board, locating pieces, generating available moves, and testing for various pieces of game state (e.g., a player's turn, whether the current player is in check, etc). Our prompts contain a short listing of the main API classes, methods, and functions available in the library. The models are instructed to generate features that receive an argument of type `chess.Board` and return a `float` value. We provide full prompts in Appendix F.

**Transformer baseline.** As a neural baseline, we train the 270M parameter Transformer architecture proposed in Ruoss et al. (2024) (their largest model) to predict the discretized win-probability for White (128 buckets) given a position encoded in the standard FEN format. Ruoss et al. (2024) compared models that predict both state values and state-action values; when controlled for the num-

ber of data points, models trained on state-value slightly outperformed when playing games against each other. Their strongest model (trained on 15.3B state-action values) achieved grandmaster-level play. Due to computational constraints, we reproduce their training of a state-value prediction Transformer run only up to 50M data points (10x less than their total state-value dataset of 500M data points; though we approximately match their number of epochs over the training data at 2.5).

Table 1 summarizes the results for this regression task. Here, we show both root mean square error (RMSE) and Pearson correlation ($\rho$) between model predictions and Stockfish's estimate. Our LeaPR models, trained on $200k$ board positions, compare favorably with the Transformer predictor trained on $250x$ more data. In contrast, as expected, Random Forests trained on the raw board struggle. LeaPR models benefit from the significant prior knowledge that LLMs have about useful chess concepts. We see basic features such as one that "Calculates the total piece value of both sides"

| Predictor | Training Size | RMSE | $\rho$ | Acc. |
|---|---|---|---|---|
| Random policy | 0 | | | 11.4% |
| Transformer | $5 \times 10^7$ | .161 | .795 | 30.3% |
| Transformer (Ruoss et al., 2024) | $5 \times 10^8$ | | | 58.5% |
| Random Forest (raw board) | 200k | .248 | .306 | 14.5% |
| PoT + GPT-5.1 | 200k | .316 | .611 | 23.10% |
| FunSearch + GPT-5.1 | 200k | .202 | .652 | 27.4% |
| **LeaPR** F2 + GPT 5-mini | $200k$ | .169 | .762 | 31.4% |
| F2 + GPT 4o-mini | $200k$ | .163 | .783 | 16.7% |
| D-ID3 + GPT 5-mini | $200k$ | .160 | .789 | 33.5% |
| D-ID3 + GPT 4o-mini | $200k$ | .156 | .806 | 17.2% |

Table 1: Performance in state-value prediction models in chess positions from Lichess. We train the 270M-parameter Transformer from Ruoss et al. (2024) with up to $50M$ data points, and report their results for their full run on $10x$ more data.

(the model's own function documentation string) proposed and implemented by GPT 4o-mini in 6 lines of code early during training, as well as significantly more complex, specific features such as "Pawn promotion pressure: sum over pawns of 1/(1+steps_to_promotion) weighted by being passed (white minus black). Encourages advanced, passed pawns.", implemented by GPT 5-mini late in the D-ID3 run (with 51 lines of code). Generally, D-ID3 features appear to become more specific as training progresses, likely because the LLM is asked to distinguish only a subset of board positions that already share many similarities (due to falling on a specific leaf node), yielding better models than F2 even with the same number of total features.

We also compare models in terms of Top-1 move accuracy compared to Stockfish at its maximum strength. Since we only estimate state-values, to use our predictors we select the move that leads to the best successor value from the point of view of the current player (i.e., the move leading to the highest or lowest win-probability for White depending on who plays), and measure how often this matches Stockfish's top move. Interestingly, we find that regression performance is not necessarily predictive of move accuracy: LeaPR models trained with GPT 4o-mini are significantly worse when used to select moves. Our best action predictors achieve non-trivial move selection accuracy: whereas random performance for this task is 11.4%, the D-ID3 model trained with GPT 5-mini predicts the top Stockfish move in 33.5% of the cases, with the Transformer

| Predictor | MNIST | Fashion |
|---|---|---|
| ResNet-50 | 98.71% | 89.54% |
| EfficientNetV2 | 98.8% | 90.94% |
| Random Forest (raw pixels) | 95.6% | 88.29% |
| PoT + GPT-5.1 | 23.75% | 22.39% |
| FunSearch + GPT-5.1 | 25.80% | 25.96% |
| **LeaPR** F2 + GPT 5 mini | 92.54% | 85.77% |
| F2 + GPT 4o mini | 89.26% | 80.26% |
| D-ID3 + GPT 5 mini | 96.91% | 88.51% |
| D-ID3 + GPT 4o mini | 93.71% | 83.80% |

Table 2: Top-1 accuracy on image classification on MNIST and Fashion-MNIST.

baseline underperforming at 30.3%.Ruoss et al. (2024) trained this same state-value model with up to 500M data points and managed to achieve a move accuracy of 58.5%, showing that the Transformer keeps improving for much longer. Their most accurate model for action prediction is trained to directly predict action-values from 15B training data points, achieving an accuracy of 63.5% and a grandmaster-level ELO rating when playing with humans online. Although there remains a significant gap between their best results achieved with a Transformer and what we demonstrate here, LeaPR models still get surprisingly far in this challenging regression task. If the scalability challenges associated with LeaPR models can be overcome (e.g., we see negligible benefits from training Random Forests beyond $200k$ training data points), we might be able to obtain chess policies that

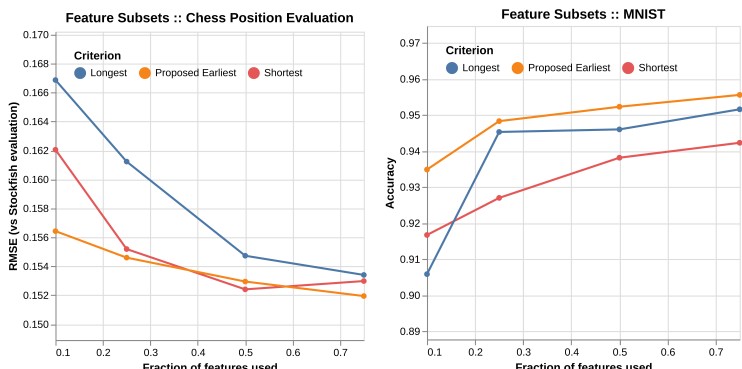

Figure 3: Random Forest predictor performance when trained on subsets of varying sizes of D-ID3-generated features (see Appendix D for remaining domains). Consistently, more D-ID3 features tends to improve the predictor's performance, and the most impactful features are found early.

not only play at a high level but can also "explain" their moves — a feature that no existing chess engine possesses.

## 4.2 Image classification

We now evaluate LeaPR models on two image classification datasets: MNIST (handwritten digit classification) from LeCun (1998) and Fashion-MNIST (grayscale fashion products classification) from Xiao et al. (2017). We train standard ResNet-50 and EfficientNetV2 baselines to convergence on the same datasets (training details in B.2). Unlike in Section 4.1, D-ID3 does not add *images* in the prompt when calling the LLM, but only a textual description of class labels of a set of training examples belonging to the leaf being split (e.g., "digit 0" in MNIST, or "T-shirt" in Fashion-MNIST). This is a significant limitation for this domain, since the features need to rely solely on the LLM's prior knowledge about what the described objects might look like and hypotheses about how they will be detectable in a small grayscale image. Still, the best LeaPR models in this task achieve comparable accuracy to the neural baselines, even without the ability to construct features by directly observing the training data. Again, especially with D-ID3 we observe specific features that attempt to distinguish between particular classes, such as *"Count of ink "endpoints": ink pixels with only one ink neighbor (8-connected). Loops like 0/8 have few endpoints; open strokes like 5 have endpoints"* being the feature with the best split (thus selected) in a leaf where the majority classes were 8 and 5. GPT 5-mini correctly implemented the above feature in 30 lines of Python using `numpy`. Though MNIST and Fashion-MNIST generally only serve as "sanity checks" for computer vision models (with even Random Forests trained on the raw pixels performing near the neural baselines, given the small image and training set sizes), we believe that this result presents an encouraging signal towards the ability of LeaPR models to achieve comparable accuracy while proposing simple and interpretable programmatic features.

## 4.3 Text classification

We now evaluate LeaPR models on a binary text classification task: detecting whether the input was written by an LLM or a human. We use the Ghostbuster dataset (Verma et al., 2024), which contains a collection of student essays, creative writing, and news articles written by humans and by ChatGPT and Claude (Anthropic, 2024) given the same or similar prompts. In Table 3 we compare LeaPR models the Ghostbuster model and other neural baselines reported in Verma et al. (2024) for the "in distribution" setting with all domains combined (we omit DetectGPT, which achieves a low F1 score of 51.6% due to having been trained to detect another LLM). Here, LeaPR models are the only neural network-free predictors. Still, our models perform competitively when evaluated in F1 score: they outperform or match all baselines, with the best LeaPR model (with features obtained by D-ID3 with GPT 5-Mini) closely matching Ghostbuster (98.8 vs 99.0 in F1 score).

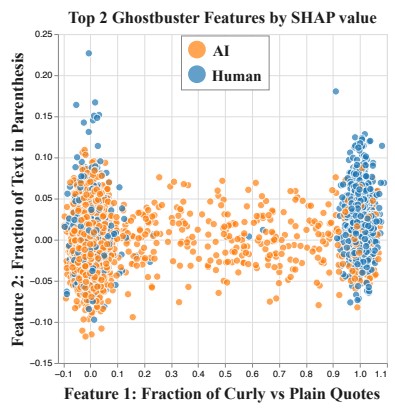

**Top 3 features by SHAP value:**

1. [0.044] Mean approximate saturation in the center region (colorfulness)
2. [0.042] Ratio of mean blue to mean green in the central region (blue vs **vegetation** center bias)
3. [0.030] - Fraction of center-region pixels that are green-dominant and have noticeable chroma (**vegetation** patches)

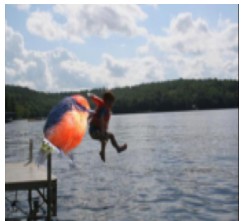

**True Label:** land bird
**Prediction:** water bird

Figure 4: (Left) Distribution of the top 2 LeaPR-learned features with highest SHAP values on the Ghostbuster dataset (gaussian jitter added to aid visualization). On Feature 1 (which measures fraction of "curly", or typographic characters, versus plain ASCII quotes), human text tends to cluster at the extremes, while AI-generated text features mid-range values. Feature 2 shows high values primarily for human-written text. (Right) A land bird misclassified by a LeaPR model on the Waterbird dataset; the top SHAP-valued features for this example show a clear reliance on the background.

## 4.4 IMPACT OF FEATURE COMPLEXITY AND QUANTITY

We now analyze the impact of both the quantity of features generated, as well as their complexity, in the quality of predictors we obtain. Here we look at D-ID3 paired with GPT-5 Mini, our best performing combination across domains. Ideally, we would like D-ID3 to steadily improve the trained predictor with a larger budget, rather than saturating early. Moreover, we would expect to obtain the most impactful features earlier, so as to make the most out of a smaller run budget.

Figure 3 shows the Random Forest predictor's performance on MNIST and Chess when trained on varying subsets of the learned features selected according to three criteria: features generated *earliest*, the longest (in lines of code), and the shortest. (Results for Fashion-MNIST and Ghostbuster, showing similar trends, are in Appendix D. Here, lines of code serves as a crude proxy for feature complexity. Across domains, we observe that (1) using more of the D-ID3-proposed features consistently improves performance, and (2) D-ID3 tends to find the most impactful features earlier: using the first features is generally better than using either the simplest (less lines of code) or most complex. Whether feature complexity is generally positive depends on the domain: for instance, in chess, simpler features seem to have a higher impact, which does not hold in the image domains. Overall, however, we see a positive scaling trend with respect to features: up to the scale of our runs, we still see improving performance the larger the budget we allow D-ID3 to use.

| Predictor | F1 |
|---|---|
| Perplexity only | 81.5 |
| GPTZero | 93.1 |
| RoBERTa | 98.1 |
| Ghostbuster | 99.0 |
| PoT + GPT-5.1 | 56.9 |
| FunSearch + GPT-5.1 | 74.2 |
| **LeaPR** F2 + GPT 5-mini | 97.7 |
| F2 + GPT 4o-mini | 98.6 |
| D-ID3 + GPT 5-mini | 98.8 |
| D-ID3 + GPT 4o-mini | 98.6 |

Table 3: F1 score on Ghostbuster: classifying text as human or AI written (Verma et al., 2024).

## 4.5 CASE STUDIES: UNDERSTANDING FEATURES AND MODEL PREDICTIONS

The interpretable representations that LeaPR models learn have potential uses beyond their predictive power. We now describe two case studies using SHAP values (Lundberg & Lee, 2017) as a lens into the patterns that LeaPR models find in their training data, and into why a particular prediction — especially if erroneous — was made. SHAP values are a metric of feature importance for a model that can be applied both at a dataset-level or to a particular prediction; we refer to Lundberg & Lee (2017) for details. This is an especially compelling tool for understanding LeaPR models given that their features already come with natural language descriptions, in the form of documentation strings.

First, we compute SHAP values in the Ghostbuster training set to understand what features the LeaPR model trained with D-ID3 and GPT 5-mini has learned to use to identify AI-generated text. Sorting features by their SHAP values on a sample of 150 training examples, we find that two of the top-3 most important features for the model are (1) the "Fraction of quotation marks that are curly/typographic quotes (e.g., ' ' " ") vs plain ASCII quotes, indicating published/edited text" and the (2) *"Proportion of characters that lie inside parentheses (measures parenthetical/planned content like "(50 words)")"* (the other top-3 feature also looks for kinds of quotation characters and is strongly correlated with the first). Together, these two features already capture distinctive patterns in human- and AI-written samples in Ghostbuster. Figure 4 (left) shows training samples projected on these two features, with small Gaussian jitter added to both coordinates to aid visualization. For feature 1, human-generated text either has value $1.0$, meaning *all* quote characters are typographical, or "curly" quotes, or $0.0$, meaning *all* quotes in the text are plain ASCII quotes (this could, for instance, reflect default settings in the user device, with using curly quotes being much more frequent). This is in stark contrast with AI-generated text, which often mixes both kinds of characters in the same text: AI-generated text displays the full range from $0$ to $1$ in this feature. For Feature 2, humans seem much more likely to wrap a significant fraction of the text in parentheses: almost all samples with value over $0.1$ in this feature (over 10% in parenthesis) were human-written. LeaPR models allowed us to quickly discover these patterns without having to formulate specific a priori assumptions: combined, our models contain thousands of automatically generated domain-relevant features, thus serving as a highly useful tool for data understanding.

Finally, we conduct a case study on the Waterbird dataset (Sagawa et al., 2020) showing how programmatic features can serve to debug model failures. This dataset contains images with two classes of birds: land birds and water birds — which are the two target classification labels. However, a trained classifier might learn instead to rely on the background, rather than use features of the bird itself. The dataset contains a subset of land birds placed on water backgrounds, and vice-versa: typically, classification accuracy drops significantly across groups when models learn to predict bird classes based on the (spuriously correlated) background. When we train a LeaPR model with F2 and GPT 5-mini on Waterbird, it achieves 100% validation accuracy when evaluated on *land birds on land background*, but it drops to 84% when evaluated on *land birds on water background*. Again, SHAP values can help us understand why this happens in a particular case. Figure 4 (right) shows the first validation example of a land bird on water background that is misclassified. When we show the top 3 features by their SHAP values, the second and third features explicitly indicate that their goal is to detect *vegetation* — a spurious feature. We can indeed find several examples of features that *attempt* to characterize the bird, such as "Fraction of warm (red-dominant) pixels in the center region (bird color cue)", that are however ignored by the trained predictor. This example shows how LeaPR models can make their failures transparent. Since model failures often reflect properties of the training data, we believe that LeaPR can serve as debugging tool for both models and datasets, applicable to a wide range of domains.

## 5 LIMITATIONS AND CONCLUSION

We introduced *Learned Programmatic Representation* models, a class of neural network-free models that combine programmatic LLM-generated features and decision tree predictors. We experiment with chess boards, images and text, seeing encouraging initial results exploring this class of models. Our learned features tend to be easy to interpret: we explore various examples across each of these domains, including using SHAP values to explain individual predictions and overall learned models.

However, several limitations remain to be tackled by future work. Our methods do not learn deep hierarchical features: each feature is directly computed from the input. Learning deep feature hierarchies is the main advantage of training *deep* neural networks, and we believe that to be an important capability for LeaPR models to perform well in more complex domains. Moreover, our experiments were done at a small scale, and there are scalability challenges — both in representation learning as well as in training predictors — to be overcome to achieve competitive performance in data-rich domains, like chess, where neural networks improve predictably with more data and compute.

Despite these limitations, we find our results encouraging for further exploring novel learning paradigms that yield interpretable models *by construction*. With a rapidly advancing AI toolbox, future tools might allow us to learn interpretable models just as easily as we can train neural networks today, with little to no sacrifice in quality. Overcoming the limitations in the LeaPR paradigm can thus be a path to incorporating interpretable models into the practical AI toolbox.

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

# A  IMPLEMENTATION DETAILS FOR F2 AND D-ID3

In this section, we first expand on the description of Algorithms 1 and 2 by detailing the subroutines they rely on, and then discuss practical implementation details of both.

## A.1  ALGORITHM SUBROUTINES

**TrainRandomForest**($D, F$): Computes feature vectors for all examples in $D$ by evaluating each $f \in F$ (parallelized via `ProcessPoolExecutor`), then trains a `scikit-learn` `RandomForestRegressor` or `RandomForestClassifier`.

**FeatureImportances**($rf$): Returns $rf$.`feature_importances`, the mean decrease in impurity for each feature.

**TopKFeatures**($F, imp$): Sorts pairs $(f, imp[f])$ for $f \in F$ by importance scores (descending) and returns the top $k$ pairs as tuples.

**RandomKFeatures**($F, imp$): Uniformly samples $k$ features from $F$ using `random.sample` (no importance weighting).

**ProposeFeatures**($LM$, context): Constructs a domain-specific prompt with context (feature examples + importances for F2; decision path + example datapoints for D-ID3), calls $LM$ API, parses response to extract Python function definitions, validates each in separate process on test subset (10k examples), and returns only those that execute without errors/timeouts/non-finite values.

**TotalError**($T, D$): For leaf node, returns domain-specific leaf error (MAE for regression or error rate for classification); for internal node, returns weighted average of children's total errors.

**SplitError**($f, t, D$): Partitions $D$ by threshold $t$ on feature $f$, computes impurity of each partition (variance for regression via `RunningMedianAbs`, entropy for classification via histogram), and returns weighted sum.

## A.2  PRACTICAL CONSIDERATIONS

Both F2 and D-ID3 run for a user-specified number of iterations; this number is exactly equal to the number of LLM calls that the algorithm will perform, allowing users to budget for LLM usage. In a sense, both algorithms are "anytime algorithms" — they can always return their latest set of learned features. The algorithms return a *representation*, rather than a predictor (e.g. the decision tree constructed by D-ID3), to allow for separation of concerns: having a representation, users can later iterate on learning predictors (which need not be decision trees) without additional LLM calls. Most of the time in F2 and D-ID3 is generally spent computing features; luckily, feature computation for all relevant examples is embarrassingly parallel, and we exploit this in our implementation. During training, we always validate proposed features on a subset of the training set (we use $10k$ examples in our experiments), and discard features that throw exceptions, timeout, or return non-finite values for some example (e.g., `NaN` or $\pm\infty$).

Our training runs for D-ID3 were the most expensive, with cost ranging from $0.5$ to $5$ US dollars per run with GPT 5-mini (1000 iterations). Runs with F2 were around 10x cheaper, due to performing 10x less LLM calls. Runs took from $5$ to $24$h on a CPU-only commodity machine.

# B  EXPERIMENTAL DETAILS

## B.1  TRANSFORMER TRAINING

We followed the architectural hyperparameters in Ruoss et al. (2024), and trained the 270M model with cross-entropy loss on their same tokenization scheme, and the same learning rate of $10^{-5}$. We train for $300k$ steps with a batch size of 400 on a machine with 4x H100 NVIDIA GPUs. This gives around 2.4 epochs over $50M$ data points. We find training to be generally stable, with top-1 move accuracy slowly but monotonically improving across checkpoints (whereas regression metrics, like Pearson correlation, seem less stable, often temporarily decreasing before improving again).

## B.2 MNIST AND FASHION-MNIST TRAINING

We train standard ResNet-50 and EfficientNet-V2 models for 4000 steps and a batch size of 1024 with Adam on a single H100 GPU. We use the default random initialization from PyTorch. For ResNet-50, we use a learning rate of 0.03; for EfficientNet-V2, we use 0.001, which we tuned using the validation set.

## B.3 FUNSEARCH AND PROGRAM OF THOUGHTS BASELINES

For our FunSearch (Novikov et al., 2025) baseline, we based our implementation on Google Deep-Mind's (https://github.com/google-deepmind/funsearch/) while adapting it to our domain. We used GPT-5.1 as the underlying base model – a much stronger model than what we used with LeaPR (GPT-5 mini). In FunSearch, we evolved 4 "islands" for 50 iterations: each island is a collection of programs. Each program is a complete task-specific predictor: for instance, in Ghostbuster, the program takes a string and outputs a class label (human or AI-generated). Unlike in the original FunSearch paper, for fairness we included the same task description and API reference that we give in our LeaPR prompts (FunSearch used a domain-agnostic prompt instead). The programs are scored during evolution in the same training set we used for LeaPR, and the best program at the end is evaluated in the test set, and its performance reported in our main results tables.

For our Program of Thoughts (Chen et al., 2022) baseline, we also gave GPT-5.1 the same task and API description as for FunSearch and LeaPR. However, here there is no evolution: we simply sample 64 full predictors from the model (divided as 8 predictors).

# C ADDITIONAL RESULTS

We conducted two preliminary experiments on additional datasets to demonstrate: (1) how LeaPR-discovered features compare with domain expert features, and (2) a limitation regarding appropriate library selection. The first experiment was on audio classification, and the second on natural language inference, a more semantic task compared to Ghostbuster. For both experiments, we evaluated D-ID3 with GPT-5-mini, using the same configurations as our main experiments.

## C.1 AUDIO CLASSIFICATION ON ESC-50

The ESC-50 dataset (Piczak) contains $2,000$ 5-second environmental audio recordings across 50 classes, including animal sounds, natural soundscapes, and urban noises.

This dataset showcases LeaPR working on yet another input modality (raw audio), but mainly it allows us to compare LeaPR's features with existing expert-generated features, since the original ESC-50 dataset paper included a baseline Random Forest trained on standard audio features. Note that, for audio, the input is very high-dimensional (at 44100 kHz for 5 seconds, each input in ESC-50 is 220,500-dimensional). The standard random forest baseline, trained on hand-designed features for audio, achieved 44.30% top-1 accuracy on ESC-50. When training LeaPR with the same configurations as in our main experiments, and with D-ID3 and gpt-5-mini, we achieve an accuracy of 64.1% with a trained random forest classifier on top. Both underperfomr the state-of-the-art neural networks, which generally achieve over 90% accuracy on ESC-50 by leveaging large-scale audio pre-training beyond ESC-50 (which only has 2000 audio samples in total). However, this allows us to directly compare LeaPR-learned features with hand-designed ones, showing a case where LeaPR improves on top of human-written features crafted for the same domain.

## C.2 SNLI

The Stanford Natural Language Inference (SNLI) dataset (Bowman et al., 2015) consists of 570k sentence pairs labeled as entailment, contradiction, or neutral. Although this is a text classification task like Ghostbuster, it requires deeper semantic understanding of the text: thus we expect the same LeaPR setup that worked well on Ghostbuster — using only the Python standard library

for manipulating strings — to work less well for tasks requiring semantics. Indeed, we find that D-ID3 with gpt-5-mini achieves an accuracy of 66.3% on SNLI, generally underperforming baselines based on fine-tuning pre-trained Transformers (which achieve 90%+ accuracy). In this setting, LeaPR features cannot reliably perform useful operations like checking if two words are synonyms or antonyms, which are highly useful for this task. We hypothesize that having access to better libraries, such as NLTK or spaCy, could be significantly beneficial here for LeaPR. Overall, this study shows that LeaPR is limited by the libraries it has access to, and it should not be expected to perform well in domains where good libraries are not available to the model.

## D    ADDITIONAL RESULTS ON FEATURE SUBSETS

Figure 5 shows the same analyses we discussed in Section 4.4 for the remaining domains. Again, features proposed earliest tend to be most impactful, with performance consistently improving with more features.

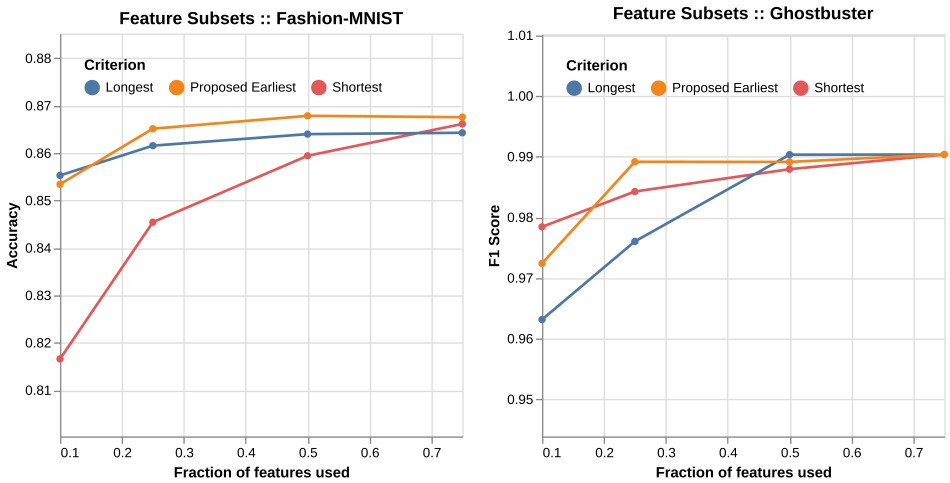

Figure 5: Same experiment from Figure 3 on Fashion-MNIST and Ghostbuster.

## E    AUTOMATED FEATURE GENERATION METHODS FOR TABULAR DATA

**LLM-based Feature Generation for Tabular Data**    Recently, OCTree (Nam et al., 2024) proposed using LLMs to generate *transformations of existing features* for *tabular data* by leveraging decision tree reasoning, where the LLM synthesizes features specifically designed to improve splits at nodes based on existing tree paths. While their approach shares our motivation of creating interpretable programmatic features through LLM synthesis, it differs in two key aspects: (1) OCTree operates on *structured tabular data with pre-existing features*, while LeaPR is more general and works with raw inputs like images or text, and (2) OCTree uses the *structure* of already-trained trees to guide feature generation, whereas our D-ID3 algorithm generates features during tree construction based on a subset of training examples that reached that node and our FunSearch variant uses evolutionary selection independent of tree structure.

Similarly to OCTree, CAAFE (Hollmann et al., 2023) also uses LLMs to generate *feature transformations* for *tabular datasets* by leveraging dataset metadata and column descriptions, then iteratively selects the most promising features through rounds of model training and feature importance analysis. While our FunSearch variant also follows a generate-then-select pipeline, it differs from both

OCTree and CAAFE in generating features from raw inputs rather than tabular data with pre-existing features.

**Rule-based Automated Feature Engineering for Tabular Data**    Alternative approaches to LLM synthesis use deterministic, rule-base methods for automated feature engineering. OpenFE (Zhang et al., 2023) proposes a two-stage approach for *tabular data*: it generates candidate features through combinatorial expansion of predefined operators (arithmetic operations, group-by aggregations, etc.) organized as operator trees, then prunes candidates using gradient boosting feature importance. While OpenFE shares our FunSearch variant's generate-and-select approach, it uses deterministic combinatorial search over a fixed operator vocabulary rather than LLM-based synthesis, limiting it to predefined transformations.

Similarly, `autofeat` (Horn et al., 2020) systematically generates polynomial and non-linear transformations of existing *tabular features* (products, ratios, logarithms) up to a specified complexity, then selects useful features via Lasso regression. While both `autofeat` and OpenFE produce interpretable mathematical expressions and achieve strong performance on *tabular benchmarks*, they differ fundamentally from our LLM-based approach in relying on *predefined transformation rules*. This limits them to standard mathematical operations and tabular data with pre-existing features, while LeaPR's LLM-based generation enables domain-specific reasoning and works directly from raw inputs across diverse modalities.

# F  PROMPTS AND EXAMPLE FEATURES

## F.1  CHESS POSITION EVALUATION

### F.1.1  F2

```
1  You are an expert chess programmer creating feature functions to help a machine learning model
       predict chess position evaluations.
2
3  Your task is to write a feature function that helps discriminate between board positions with
       different evaluations (e.g., probability that white or black wins).
4  A feature function is a Python function that takes a chess Board and computes a feature out of
       the board. It should return a float, but note that a feature could also be effectively
       boolean-valued (0.0 or 1.0), or integer-valued, even if its type is float.
5
6  You have access to the following API from the `chess` library:
7
8  # Chess API Documentation
9  ## Class chess.Board Methods
10 - board.turn: True if White to move, False if Black
11 - board.fullmove_number: Current move number
12 - board.halfmove_clock: Halfmove clock for 50-move rule
13 - board.is_check(): True if current player is in check
14 - board.is_checkmate(): True if current player is checkmated
15 - board.is_stalemate(): True if stalemate
16 - board.is_insufficient_material(): Returns True if insufficient material
17 - board.piece_at(square): Returns piece at given square (or None)
18 - board.piece_map(): Returns dict mapping squares to pieces
19 - board.legal_moves: Iterator over legal moves
20 - board.attackers(color, square): Returns set of squares attacking given square
21 - board.is_attacked_by(color, square): Returns True if square is attacked by color
22
23 ## Chess Squares and Pieces
24 - chess.A1, chess.A2, ..., chess.H8: Square constants
25 - chess.PAWN, chess.KNIGHT, chess.BISHOP, chess.ROOK, chess.QUEEN, chess.KING: Piece types
26 - chess.WHITE, chess.BLACK: Colors
27 - piece.piece_type: Type of piece (PAWN, KNIGHT, etc.)
28 - piece.color: Color of piece (WHITE or BLACK)
29
30 ## Useful Functions
31 - chess.square_name(square): Convert square index to name (e.g., "e4")
32 - chess.parse_square(name): Convert square name to index
33 - chess.square_file(square): Get file (0-7) of square
34 - chess.square_rank(square): Get rank (0-7) of square
35 - chess.square_distance(sq1, sq2): Manhattan distance between squares
36
37
38 ## Current Feature Database
```

```
39  Here are some of our existing features and their importance to the current model (higher
        importance means this is a more useful feature for the current model):
40
41  Feature:
42  def feature(board: chess.Board) -> float:
43      'King centralization in endgames: positive if White king is closer to center than Black
         king (only active when low material)'
44      total_non_king_pieces = sum(1 for _, p in board.piece_map().items() if p.piece_type !=
         chess.KING)
45      # Activation threshold: small material (pawns + pieces <= 6)
46      if total_non_king_pieces > 6:
47          return 0.0
48      center_sq = [chess.parse_square(s) for s in ('d4','e4','d5','e5')]
49      center_sq_avg = sum(center_sq)  # not used directly; use center coords (3.5,3.5)
50      def king_dist(color):
51          for sq, p in board.piece_map().items():
52              if p.piece_type == chess.KING and p.color == color:
53                  # distance to center by min over central squares
54                  return min(chess.square_distance(sq, c) for c in center_sq)
55          return 8.0
56      wd = king_dist(chess.WHITE)
57      bd = king_dist(chess.BLACK)
58      # Normalize in range roughly -1..1
59      return float((bd - wd) / 8.0)
60
61
62  Importance: 0.014
63  ---
64
65
66  Feature:
67  def feature(board: chess.Board) -> float:
68      'Bishop-pair and minor-piece balance: (White advantage) bishops and minor piece
         composition bonus'
69      w_bishops = 0
70      b_bishops = 0
71      w_minors = 0
72      b_minors = 0
73      for _, p in board.piece_map().items():
74          if p.piece_type == chess.BISHOP:
75              if p.color == chess.WHITE:
76                  w_bishops += 1
77              else:
78                  b_bishops += 1
79          if p.piece_type in (chess.BISHOP, chess.KNIGHT):
80              if p.color == chess.WHITE:
81                  w_minors += 1
82              else:
83                  b_minors += 1
84      score = 0.0
85      # bishop pair bonus
86      if w_bishops >= 2:
87          score += 0.6
88      if b_bishops >= 2:
89          score -= 0.6
90      # minor piece imbalance small weight
91      score += 0.12 * (w_minors - b_minors)
92      return float(score)
93
94
95  Importance: 0.039
96  ---
97
98
99  Feature:
100 def feature(board: chess.Board) -> float:
101     'Center control: difference in control of d4,e4,d5,e5 (occupied=1, attacked=0.5)'
102     center_squares = [chess.parse_square(s) for s in ('d4','e4','d5','e5')]
103     def control_for(color):
104         c = 0.0
105         for sq in center_squares:
106             occ = board.piece_at(sq)
107             if occ is not None and occ.color == color:
108                 c += 1.0
109             # attacked by color
110             if board.is_attacked_by(color, sq):
111                 c += 0.5
112         return c
113     wc = control_for(chess.WHITE)
114     bc = control_for(chess.BLACK)
115     return float(wc - bc)
```

```
116
117
118  Importance: 0.044
119  ---
120
121
122  Feature:
123  def feature(board: chess.Board) -> float:
124      'Piece activity squares: difference in number of unique squares attacked by non-pawn, non-
          king pieces (White - Black) normalized by 64'
125      def active_squares(color):
126          count = 0
127          for sq in range(64):
128              attackers = board.attackers(color, sq)
129              found = False
130              for a in attackers:
131                  p = board.piece_at(a)
132                  if p is None:
133                      continue
134                  if p.piece_type not in (chess.PAWN, chess.KING):
135                      found = True
136                      break
137              if found:
138                  count += 1
139          return count
140      w = active_squares(chess.WHITE)
141      b = active_squares(chess.BLACK)
142      return float((w - b) / 64.0)
143
144
145  Importance: 0.329
146  ---
147
148
149  Feature:
150  def feature(board: chess.Board) -> float:
151      'Pawn structure weakness: (Black penalties - White penalties), positive if Black worse (
          good for White). Penalty = doubled*0.5 + isolated*0.7'
152      def pawn_penalty(color):
153          files = {f:0 for f in range(8)}
154          pawn_sqs = []
155          for sq, p in board.piece_map().items():
156              if p.piece_type == chess.PAWN and p.color == color:
157                  f = chess.square_file(sq)
158                  files[f] += 1
159                  pawn_sqs.append(sq)
160          doubled = sum(max(0, cnt-1) for cnt in files.values())
161          isolated = 0
162          for sq in pawn_sqs:
163              f = chess.square_file(sq)
164              if files.get(f-1,0) == 0 and files.get(f+1,0) == 0:
165                  isolated += 1
166          return doubled * 0.5 + isolated * 0.7
167      bp = pawn_penalty(chess.BLACK)
168      wp = pawn_penalty(chess.WHITE)
169      return float(bp - wp)
170
171
172  Importance: 0.065
173  ---
174
175
176  Feature:
177  def feature(board: chess.Board) -> float:
178      'King safety pressure: weighted sum of attackers near each king (positive = pressure on
          Black king > White king)'
179      def king_square(color):
180          for sq, p in board.piece_map().items():
181              if p.piece_type == chess.KING and p.color == color:
182                  return sq
183          return None
184      wk = king_square(chess.WHITE)
185      bk = king_square(chess.BLACK)
186      values = {chess.PAWN:1.0, chess.KNIGHT:3.0, chess.BISHOP:3.25, chess.ROOK:5.0, chess.QUEEN
          :9.0, chess.KING:0.5}
187      def pressure_on(king_sq, attacker_color):
188          if king_sq is None:
189              return 0.0
190          attackers = board.attackers(attacker_color, king_sq)
191          s = 0.0
192          for a in attackers:
```

```
193            p = board.piece_at(a)
194            if p is None:
195                continue
196            dist = chess.square_distance(a, king_sq)
197            s += values.get(p.piece_type, 0.0) / (1.0 + dist)
198        return s
199    p_on_black = pressure_on(bk, chess.WHITE)
200    p_on_white = pressure_on(wk, chess.BLACK)
201    return float(p_on_black - p_on_white)
202
203
204 Importance: 0.005
205 ---
206
207
208 Feature:
209 def feature(board: chess.Board) -> float:
210     'Undefended high-value threats: (value of Black pieces under more attackers than defenders
        ) - (value of White pieces similarly threatened)'
211     values = {chess.PAWN:1.0, chess.KNIGHT:3.0, chess.BISHOP:3.25, chess.ROOK:5.0, chess.QUEEN
        :9.0, chess.KING:0.0}
212     threat_black = 0.0
213     threat_white = 0.0
214     for sq, p in board.piece_map().items():
215         if p.piece_type == chess.PAWN:
216             continue
217         attackers = board.attackers(not p.color, sq)
218         defenders = board.attackers(p.color, sq)
219         atk = sum(1 for _ in attackers)
220         defn = sum(1 for _ in defenders)
221         if atk > defn and atk > 0:
222             score = values.get(p.piece_type, 0.0) * (atk - defn)
223             if p.color == chess.BLACK:
224                 threat_black += score
225             else:
226                 threat_white += score
227     return float(threat_black - threat_white)
228
229 Importance: 0.045
230 ---
231
232
233 Feature:
234 def feature(board: chess.Board) -> float:
235     'Material balance (White minus Black) using common piece values: P=1,N=3,B=3.25,R=5,Q=9'
236     values = {chess.PAWN:1.0, chess.KNIGHT:3.0, chess.BISHOP:3.25, chess.ROOK:5.0, chess.QUEEN
        :9.0, chess.KING:0.0}
237     total = 0.0
238     for sq, piece in board.piece_map().items():
239         val = values.get(piece.piece_type, 0.0)
240         total += val if piece.color == chess.WHITE else -val
241     return float(total)
242
243
244 Importance: 0.199
245 ---
246
247
248 Feature:
249 def feature(board: chess.Board) -> float:
250     'Normalized mobility difference: (White legal moves - Black legal moves) / 100'
251     try:
252         white_moves = 0
253         black_moves = 0
254         # count current side moves
255         white_board = board.copy()
256         white_board.turn = chess.WHITE
257         white_moves = sum(1 for _ in white_board.legal_moves)
258         black_board = board.copy()
259         black_board.turn = chess.BLACK
260         black_moves = sum(1 for _ in black_board.legal_moves)
261         return float((white_moves - black_moves) / 100.0)
262     except Exception:
263         return 0.0
264
265
266 Importance: 0.165
267 ---
268
269
270 Feature:
```

```
271 def feature(board: chess.Board) -> float:
272     'Passed pawns score: sum of passed-pawn strengths (White minus Black), advanced pawns
        weighted more'
273     def is_passed(sq, color):
274         f = chess.square_file(sq)
275         r = chess.square_rank(sq)
276         if color == chess.WHITE:
277             ahead_ranks = range(r+1, 8)
278             opp_color = chess.BLACK
279             for ar in ahead_ranks:
280                 for df in (-1,0,1):
281                     ff = f + df
282                     if 0 <= ff < 8:
283                         sq2 = chess.square(ff, ar)
284                         p = board.piece_at(sq2)
285                         if p is not None and p.color == opp_color and p.piece_type == chess.
        PAWN:
286                             return False
287             return True
288         else:
289             ahead_ranks = range(r-1, -1, -1)
290             opp_color = chess.WHITE
291             for ar in ahead_ranks:
292                 for df in (-1,0,1):
293                     ff = f + df
294                     if 0 <= ff < 8:
295                         sq2 = chess.square(ff, ar)
296                         p = board.piece_at(sq2)
297                         if p is not None and p.color == opp_color and p.piece_type == chess.
        PAWN:
298                             return False
299             return True
300     score_w = 0.0
301     score_b = 0.0
302     for sq, p in board.piece_map().items():
303         if p.piece_type != chess.PAWN:
304             continue
305         rank = chess.square_rank(sq)
306         if p.color == chess.WHITE:
307             if is_passed(sq, chess.WHITE):
308                 advancement = (rank - 1) / 6.0 if rank >= 1 else 0.0
309                 score_w += 1.0 + advancement
310         else:
311             if is_passed(sq, chess.BLACK):
312                 advancement = (6 - rank) / 6.0 if rank <= 6 else 0.0
313                 score_b += 1.0 + advancement
314     return float(score_w - score_b)
315
316
317 Importance: 0.094
318 ---
319
320
321 # Task
322 Generate 10 new chess board feature functions in Python that:
323
324 1. Help us discriminate between strong and weak board positions, hopefully with positions
        before and after the optimal split point having the lowest possible variance between
        their evaluations.
325 2. Return a float value given a board position.
326 3. Handle edge cases gracefully - won't crash on unusual positions
327
328 Your task is to generate diverse, creative features that are relevant to explain the
        evaluations for the board positions shown above. Focus on features that would help
        distinguish between positions of different strengths. These features will be used in this
        decision tree that will predict the evaluation of a given board position in estimated %
        win probability for white (e.g., 20 means Black winning with around 80% probability).
        Think about new features that would help such a predictor in the particular cases above,
        trying to add information that the already existing features shown above are missing.
329
330 # Code Requirements
331
332 - Use single quotes for docstrings: "description here"
333 - No markdown code blocks
334 - No explanatory text after the function
335 - Each function should be complete and standalone, and return a float
336
337 # Output Format
338 Generate exactly 10 features in this format:
339
340 def feature(board: chess.Board) -> float:
```

```
341     "Simple, clear description of what this feature measures"
342     # ... Calculate and return the feature value
343     return result
344
345 def feature(board: chess.Board) -> float:
346     "Another feature description"
347     # ... Calculate and return the feature value
348     return result
349
350 The body of the function can be anything, but the first line (function declaration) should be
        identical to those examples above, and the second line should be a one-line docstring.
        Don't output explanatory text - just the function definitions as shown above.
351
352 Optimize for producing discriminant features that are novel compared to the existing features
        and that are likely to achieve a high importance for scoring positions, once we retrain
        the model using your new features combined with the existing ones.
```

### F.1.2  D-ID3 - PROMPT

```
1 You are an expert chess programmer creating feature functions to help a machine learning model
        predict chess position evaluations.
2
3 Your task is to write a feature function that helps discriminate between the board positions
        given below.
4 A feature function is a Python function that takes a chess Board and computes a feature out of
        the board. It should return a float, but note that a feature could also be effectively
        boolean-valued (0.0 or 1.0), or integer-valued, even if its type is float.
5
6 You have access to the following API from the 'chess' library:
7
8 # Chess API Documentation
9 ## Class chess.Board Methods
10 - board.turn: True if White to move, False if Black
11 - board.fullmove_number: Current move number
12 - board.halfmove_clock: Halfmove clock for 50-move rule
13 - board.is_check(): True if current player is in check
14 - board.is_checkmate(): True if current player is checkmated
15 - board.is_stalemate(): True if stalemate
16 - board.is_insufficient_material(): Returns True if insufficient material
17 - board.piece_at(square): Returns piece at given square (or None)
18 - board.piece_map(): Returns dict mapping squares to pieces
19 - board.legal_moves: Iterator over legal moves
20 - board.attackers(color, square): Returns set of squares attacking given square
21 - board.is_attacked_by(color, square): Returns True if square is attacked by color
22
23 ## Chess Squares and Pieces
24 - chess.A1, chess.A2, ..., chess.H8: Square constants
25 - chess.PAWN, chess.KNIGHT, chess.BISHOP, chess.ROOK, chess.QUEEN, chess.KING: Piece types
26 - chess.WHITE, chess.BLACK: Colors
27 - piece.piece_type: Type of piece (PAWN, KNIGHT, etc.)
28 - piece.color: Color of piece (WHITE or BLACK)
29
30 ## Useful Functions
31 - chess.square_name(square): Convert square index to name (e.g., "e4")
32 - chess.parse_square(name): Convert square name to index
33 - chess.square_file(square): Get file (0-7) of square
34 - chess.square_rank(square): Get rank (0-7) of square
35 - chess.square_distance(sq1, sq2): Manhattan distance between squares
36
37
38 # Task
39 Generate 10 new chess board feature functions in Python that:
40
41 1. Help us discriminate between strong and weak board positions, hopefully with positions
        before and after the optimal split point having the lowest possible variance between
        their evaluations.
42 2. Return a float value given a board position.
43 3. Handle edge cases gracefully - won't crash on unusual positions
44
45 Your task is to generate diverse, creative features that are relevant to explain the
        evaluations for the board positions shown above. Focus on features that would help
        distinguish between positions of different strengths. These features will be used in this
         decision tree that will predict the evaluation of a given board position in estimated %
        win probability for white (e.g., 20 means Black winning with around 80% probability).
        Think about new features that would help such a predictor in the particular cases above,
        trying to add information that the already existing features shown above are missing.
46
47 # Code Requirements
48
49 - Use single quotes for docstrings: "description here"
```

```
50  - No markdown code blocks
51  - No explanatory text after the function
52  - Each function should be complete and standalone, and return a float
53
54  # Output Format
55  Generate exactly 10 features in this format:
56
57  def feature(board: chess.Board) -> float:
58      "Simple, clear description of what this feature measures"
59      # ... Calculate and return the feature value
60      return result
61
62  def feature(board: chess.Board) -> float:
63      "Another feature description"
64      # ... Calculate and return the feature value
65      return result
66
67  The body of the function can be anything, but the first line (function declaration) should be
        identical to those examples above, and the second line should be a one-line docstring.
        Don't output explanatory text - just the function definitions as shown above.
68
69  # Current decision tree node
70  You are currently focusing on features that explain the position evaluation of board positions
        in the following subtree of a decision tree:
71
72  [root]
73   -> value < 3.000 for "Measure the material balance between both players"  -> value < 1.182
        for "Counts the number of pieces for each player and computes the ratio of the piece
        counts."  -> value > 0.650 for "Counts the number of squares attacked by pieces of each
        color to assess control of the board."
74
75  # Board positions
76  Here are examples of board positions in this subtree, along with their position evaluations (
        computed by Stockfish):
77
78  Board:
79  . . r r . . k .
80  . . . . . p p p
81  p b . . p . . .
82  . p . . P . . .
83  . P . . N P n q
84  P . . . . . P .
85  . B . . Q . . P
86  R . . . . R . K
87  Evaluation: 38.08281678856247
88  ---
89
90  Board:
91  r . . . . . k .
92  . b . r . p p .
93  p . . P . . . .
94  . p . . . . . Q
95  . P . b . . . .
96  P . . . . . . .
97  . . . . . . P P
98  . . . . . R . K
99  Evaluation: 15.383900101915987
100 ---
101
102 Board:
103 r n b . k . . r
104 p p p . n p p p
105 . . . b . . . .
106 . . . B . . . .
107 . . . P P p . q
108 . . . . . . . .
109 P P P . . . P P
110 R N B Q . K N R
111 Evaluation: 47.700323643230064
112 ---
113
114 Board:
115 . . . r . . . k
116 p p . . . p p .
117 . . . . p . . p
118 . . b r P . . P
119 . q . N R . . .
120 . . . . B . . .
121 . . . . . K P .
122 . . R Q . . . .
123 Evaluation: 50.0
```

```
124  ---
125
126  Board:
127  r n b q k b n r
128  p p p p . p p p
129  . . . . . . . .
130  . . . . p . . .
131  . . . . P . . .
132  . . . . . N . .
133  P P P P . P P P
134  R N B Q K B . R
135  Evaluation: 51.19644756166626
136  ---
137
138
139  Optimize for producing discriminant features that are novel compared to the existing features
         used to arrive at this subtree. Focus on explaining the differences between the board
         positions shown above.
```

### F.1.3  D-ID3 - EXAMPLE FEATURES

```python
def feature(board: chess.Board) -> float:
    "Measure the material balance between both players"
    material_values = {chess.PAWN: 1, chess.KNIGHT: 3, chess.BISHOP: 3, chess.ROOK: 5, chess.
     QUEEN: 9, chess.KING: 0}
    white_material = sum(material_values[piece.piece_type] for piece in board.piece_map().
     values() if piece.color == chess.WHITE)
    black_material = sum(material_values[piece.piece_type] for piece in board.piece_map().
     values() if piece.color == chess.BLACK)
    return float(white_material - black_material)

def feature(board: chess.Board) -> float:
    "Counts the number of pieces for each player and computes the ratio of the piece counts."
    piece_count_white = sum(1 for piece in board.piece_map().values() if piece.color == chess.
     WHITE)
    piece_count_black = sum(1 for piece in board.piece_map().values() if piece.color == chess.
     BLACK)
    if piece_count_black == 0:
        return float('inf')  # Black has no pieces left
    return piece_count_white / piece_count_black

def feature(board: chess.Board) -> float:
    "Counts the number of squares attacked by pieces of each color to assess control of the
     board."
    white_attacks = sum(board.is_attacked_by(chess.WHITE, square) for square in chess.SQUARES)
    black_attacks = sum(board.is_attacked_by(chess.BLACK, square) for square in chess.SQUARES)
    control_ratio = white_attacks / (black_attacks + 1)  # Avoid division by zero
    return float(control_ratio)

def feature(board: chess.Board) -> float:
    "Calculates the center control by counting pieces in the central squares."
    central_squares = [chess.D4, chess.D5, chess.E4, chess.E5]
    control = sum(1 for square in central_squares if board.piece_at(square) is not None)
    return float(control)

def feature(board: chess.Board) -> float:
    "Calculates the total piece value for each player based on standard chess piece values."
    piece_values = {
        chess.PAWN: 1,
        chess.KNIGHT: 3,
        chess.BISHOP: 3,
        chess.ROOK: 5,
        chess.QUEEN: 9,
        chess.KING: 0  # King is invaluable
    }
    white_value = sum(piece_values[piece.piece_type] for piece in board.piece_map().values()
     if piece.color == chess.WHITE)
    black_value = sum(piece_values[piece.piece_type] for piece in board.piece_map().values()
     if piece.color == chess.BLACK)
    return float(white_value - black_value)
```

## F.2  IMAGE CLASSIFICATION

### F.2.1  F2

```
You are an expert computer vision programmer creating evaluation features for a machine
    learning model that predicts values from images.
```

```
3   This is a classification task with the following classes: 0: landbird, 1: waterbird.
4
5   Your task is to write a feature function that helps discriminate between the image classes
        above.
6   A feature function is a Python function that takes an image array and computes a feature out
        of the image. It should return a float, but note that a feature could also be effectively
        boolean-valued (0.0 or 1.0), or integer-valued, even if its type is float.
7
8   You have access to the following API from image processing libraries:
9
10
11  # Image Processing API Documentation
12
13  The features receive an image as a numpy array, so you can use any numpy functions on it. For
        RGB images, shape is (height, width, 3). For grayscale, shape is (height, width).
14
15  ## Image Processing Methods
16  - image.shape: Returns (height, width, channels) for RGB or (height, width) for grayscale
17  - image.mean(): Average pixel intensity across all channels
18  - image.std(): Standard deviation of pixel intensities
19  - image.max(), image.min(): Maximum and minimum pixel values
20  - np.sum(image): Sum of all pixel values
21  - np.count_nonzero(image): Count of non-zero pixels
22
23  ## Handle Both Grayscale and RGB
24  - Check format: len(image.shape) == 2 for grayscale, len(image.shape) == 3 for RGB
25  - Unpack safely: h, w = image.shape[:2]  # Works for both formats
26  - For RGB only: image[:,:,0] (red), image[:,:,1] (green), image[:,:,2] (blue)
27
28  ## Useful Functions
29  - np.mean(image): Average intensity
30  - np.std(image): Standard deviation
31  - np.gradient(image): Image gradients - for RGB use on single channel: np.gradient(image
        [:,:,0])
32  - np.where(condition, x, y): Conditional selection
33  - np.argmax(image), np.argmin(image): Location of max/min values
34  - np.percentile(image, q): Percentile values
35  - np.histogram(image.flatten(), bins): Intensity histogram
36
37  ## Spatial Analysis
38  - image[start_row:end_row, start_col:end_col]: Region selection
39  - Center region: image[h//4:3*h//4, w//4:3*w//4]
40  - Edge detection: np.gradient(np.mean(image, axis=2)) for RGB
41  - Color channel differences: image[:,:,0] - image[:,:,1]
42
43  ## Example Feature Function
44  def feature(image: np.ndarray) -> float:
45      "Average pixel intensity in the center region"
46      if len(image.shape) == 3:
47          h, w, c = image.shape
48          gray = np.mean(image, axis=2)
49      else:
50          h, w = image.shape
51          gray = image
52      center_h, center_w = h // 4, w // 4
53      center_region = gray[center_h:3*center_h, center_w:3*center_w]
54      return float(np.mean(center_region))
55
56
57  ## Current Feature Database
58  Here are some existing features and their importances to the current classifier (importance =
        benefit from that feature, higher is better):
59
60  Feature:
61  def feature(image: np.ndarray) -> float:
62      'Relative difference in average blue intensity between bottom half and top half'
63      import numpy as np
64      h, w = image.shape[:2]
65      if len(image.shape) != 3 or image.shape[2] < 3:
66          return float(0.0)
67      b = image[:, :, 2].astype(float)
68      top_mean = np.mean(b[:h // 2, :]) if h // 2 > 0 else np.mean(b)
69      bot_mean = np.mean(b[h // 2:, :]) if h - h // 2 > 0 else np.mean(b)
70      denom = (np.mean(b) + 1e-8)
71      return float((bot_mean - top_mean) / denom)
72
73
74  Importance: 0.081
75  ---
76
77
```

```
78  Feature:
79  def feature(image: np.ndarray) -> float:
80      'Aspect ratio (width/height) of bounding box of pixels significantly different from median
          intensity'
81      import numpy as np
82      h, w = image.shape[:2]
83      if len(image.shape) == 3:
84          gray = np.mean(image, axis=2).astype(float)
85      else:
86          gray = image.astype(float)
87      med = np.median(gray)
88      dynamic_range = np.max(gray) - np.min(gray)
89      thresh = 0.15 * (dynamic_range + 1e-8)
90      mask = np.abs(gray - med) > thresh
91      if not np.any(mask):
92          return float(0.5)
93      rows = np.where(np.any(mask, axis=1))[0]
94      cols = np.where(np.any(mask, axis=0))[0]
95      if rows.size == 0 or cols.size == 0:
96          return float(0.5)
97      r0, r1 = rows[0], rows[-1]
98      c0, c1 = cols[0], cols[-1]
99      bbox_h = (r1 - r0 + 1)
100     bbox_w = (c1 - c0 + 1)
101     return float(bbox_w / (bbox_h + 1e-8))
102
103 Importance: 0.082
104 ---
105
106
107 Feature:
108 def feature(image: np.ndarray) -> float:
109     'Left-right symmetry score (1.0 = perfectly symmetric, lower = less symmetric)'
110     import numpy as np
111     if len(image.shape) == 3:
112         gray = np.mean(image, axis=2).astype(float)
113     else:
114         gray = image.astype(float)
115     flipped = np.fliplr(gray)
116     diff = np.mean(np.abs(gray - flipped))
117     # normalize by image contrast to avoid small-image issues
118     denom = np.mean(np.abs(gray - np.mean(gray))) + 1e-8
119     norm_diff = diff / denom
120     score = 1.0 - np.tanh(norm_diff)  # bounded between ~0 and 1
121     return float(np.clip(score, 0.0, 1.0))
122
123
124 Importance: 0.084
125 ---
126
127
128 Feature:
129 def feature(image: np.ndarray) -> float:
130     'Ratio of average horizontal gradient magnitude to vertical gradient magnitude (texture
         orientation)'
131     import numpy as np
132     # compute gray
133     if len(image.shape) == 3:
134         gray = np.mean(image, axis=2).astype(float)
135     else:
136         gray = image.astype(float)
137     gy, gx = np.gradient(gray)
138     mean_dx = np.mean(np.abs(gx))
139     mean_dy = np.mean(np.abs(gy))
140     return float(mean_dx / (mean_dy + 1e-8))
141
142
143 Importance: 0.159
144 ---
145
146
147 Feature:
148 def feature(image: np.ndarray) -> float:
149     'Proportion of pixels where green channel is significantly high (vegetation cue)'
150     import numpy as np
151     h, w = image.shape[:2]
152     if len(image.shape) != 3 or image.shape[2] < 3:
153         return float(0.0)
154     img = image.astype(float)
155     r, g, b = img[:, :, 0], img[:, :, 1], img[:, :, 2]
156     # require green greater than both red and blue and at least above 60th percentile
```

```
157    thresh = np.percentile(g.flatten(), 60)
158    mask = (g > r) & (g > b) & (g > thresh)
159    return float(np.count_nonzero(mask) / (h * w + 1e-12))
160
161
162 Importance: 0.094
163 ---
164
165
166 Feature:
167 def feature(image: np.ndarray) -> float:
168    'Fraction of pixels brighter than the 90th percentile (bright spot proportion)'
169    import numpy as np
170    if len(image.shape) == 3:
171        gray = np.mean(image, axis=2).astype(float)
172    else:
173        gray = image.astype(float)
174    thresh = np.percentile(gray.flatten(), 90)
175    frac = np.count_nonzero(gray > thresh) / (gray.size + 1e-12)
176    return float(frac)
177
178
179 Importance: 0.080
180 ---
181
182
183 Feature:
184 def feature(image: np.ndarray) -> float:
185    'Normalized difference between center region brightness and border brightness'
186    import numpy as np
187    h, w = image.shape[:2]
188    if len(image.shape) == 3:
189        gray = np.mean(image, axis=2).astype(float)
190    else:
191        gray = image.astype(float)
192    ch0, ch1 = h // 4, w // 4
193    center = gray[ch0:3 * ch0 or h, ch1:3 * ch1 or w]
194    # border defined as whole minus center
195    mask_border = np.ones_like(gray, dtype=bool)
196    mask_border[ch0:3 * ch0 or h, ch1:3 * ch1 or w] = False
197    center_mean = np.mean(center) if center.size > 0 else 0.0
198    border_mean = np.mean(gray[mask_border]) if np.any(mask_border) else 0.0
199    denom = np.mean(np.abs(gray)) + 1e-8
200    return float((center_mean - border_mean) / denom)
201
202
203 Importance: 0.088
204 ---
205
206
207 Feature:
208 def feature(image: np.ndarray) -> float:
209    'Proportion of pixels where blue channel is dominant (blue > red and blue > green)'
210    import numpy as np
211    h, w = image.shape[:2]
212    if len(image.shape) != 3 or image.shape[2] < 3:
213        return float(0.0)
214    img = image.astype(float)
215    r, g, b = img[:, :, 0], img[:, :, 1], img[:, :, 2]
216    mask = (b > r) & (b > g)
217    return float(np.count_nonzero(mask) / (h * w + 1e-12))
218
219
220 Importance: 0.117
221 ---
222
223
224 Feature:
225 def feature(image: np.ndarray) -> float:
226    'Edge density: fraction of pixels with gradient magnitude above (mean+std)'
227    import numpy as np
228    if len(image.shape) == 3:
229        gray = np.mean(image, axis=2).astype(float)
230    else:
231        gray = image.astype(float)
232    gy, gx = np.gradient(gray)
233    mag = np.hypot(gx, gy)
234    thresh = np.mean(mag) + np.std(mag)
235    count = np.count_nonzero(mag > thresh)
236    total = gray.size
237    return float(count / (total + 1e-12))
```

```
238
239
240  Importance: 0.114
241  ---
242
243
244  Feature:
245  def feature(image: np.ndarray) -> float:
246      'Average per-pixel color "saturation" approximated by (max-min)/max across channels'
247      import numpy as np
248      if len(image.shape) != 3 or image.shape[2] < 3:
249          return float(0.0)
250      img = image.astype(float)
251      mx = np.max(img, axis=2)
252      mn = np.min(img, axis=2)
253      # avoid division by zero
254      sat = (mx - mn) / (mx + 1e-8)
255      return float(np.mean(sat))
256
257
258  Importance: 0.100
259  ---
260
261
262  ## Task
263  Generate 10 NEW image features that:
264
265  1. Are different from existing features
266  2. Capture useful visual patterns
267  3. Return float values
268  4. Handle edge cases gracefully - Won't crash on unusual images
269  5. Use simple, short docstrings - Use single quotes, not triple quotes
270  6. Are efficient to compute
271
272  Your task is to generate diverse, creative features that capture different aspects of image
         content for prediction. Focus on features that would help distinguish between different
         samples. These features will be used as input to a learned model that predicts target
         values from images.
273
274  ## IMPORTANT CODE REQUIREMENTS
275  - Use SINGLE quotes for docstrings: "description here"
276  - NO triple quotes (""") anywhere in the code
277  - NO markdown code blocks
278  - NO explanatory text after the function
279  - Each function should be complete and standalone
280
281  ## Output Format
282  Generate exactly 10 features in this format:
283
284  def feature(image: np.ndarray) -> float:
285      "Clear description of what this feature measures"
286      # ... Calculate and return the feature value
287      return float(result)
288
289  def feature(image: np.ndarray) -> float:
290      "Another feature description"
291      # ... Calculate and return the feature value
292      return float(result)
293
294  The body of the functions can be anything, but the first line (function declaration) should be
         identical to those examples above (always 'def feature(...)'), and the second line
         should be a one-line docstring. Don't output explanatory text - just the function
         definitions as shown above.
```

### F.2.2 D-ID3 - PROMPT

```
1  You are an expert image processing programmer creating feature functions to help a machine
       learning model perform image classification.
2
3  This is a classification task with the following classes: 0: digit zero, 1: digit one, 2:
       digit two, 3: digit three, 4: digit four, 5: digit five, 6: digit six, 7: digit seven, 8:
       digit eight, 9: digit nine.
4
5  Your task is to write a feature function that helps discriminate between the image samples
       given below.
6  A feature function is a Python function that takes an image array and computes a feature out
       of the image. It should return a float, but note that a feature could also be effectively
       boolean-valued (0.0 or 1.0), or integer-valued, even if its type is float.
7
8  You have access to the following API from image processing libraries:
```

```
9
10
11  # Image Processing API Documentation
12
13  The features receive an image as a numpy array, so you can use any numpy functions on it. For
        RGB images, shape is (height, width, 3). For grayscale, shape is (height, width).
14
15  ## Image Processing Methods
16  - image.shape: Returns (height, width, channels) for RGB or (height, width) for grayscale
17  - image.mean(): Average pixel intensity across all channels
18  - image.std(): Standard deviation of pixel intensities
19  - image.max(), image.min(): Maximum and minimum pixel values
20  - np.sum(image): Sum of all pixel values
21  - np.count_nonzero(image): Count of non-zero pixels
22
23  ## Handle Both Grayscale and RGB
24  - Check format: len(image.shape) == 2 for grayscale, len(image.shape) == 3 for RGB
25  - Unpack safely: h, w = image.shape[:2]  # Works for both formats
26  - For RGB only: image[:,:,0] (red), image[:,:,1] (green), image[:,:,2] (blue)
27
28  ## Useful Functions
29  - np.mean(image): Average intensity
30  - np.std(image): Standard deviation
31  - np.gradient(image): Image gradients - for RGB use on single channel: np.gradient(image
        [:,:,0])
32  - np.where(condition, x, y): Conditional selection
33  - np.argmax(image), np.argmin(image): Location of max/min values
34  - np.percentile(image, q): Percentile values
35  - np.histogram(image.flatten(), bins): Intensity histogram
36
37  ## Spatial Analysis
38  - image[start_row:end_row, start_col:end_col]: Region selection
39  - Center region: image[h//4:3*h//4, w//4:3*w//4]
40  - Edge detection: np.gradient(np.mean(image, axis=2)) for RGB
41  - Color channel differences: image[:,:,0] - image[:,:,1]
42
43  ## Example Feature Function
44  def feature(image: np.ndarray) -> float:
45      "Average pixel intensity in the center region"
46      if len(image.shape) == 3:
47          h, w, c = image.shape
48          gray = np.mean(image, axis=2)
49      else:
50          h, w = image.shape
51          gray = image
52      center_h, center_w = h // 4, w // 4
53      center_region = gray[center_h:3*center_h, center_w:3*center_w]
54      return float(np.mean(center_region))
55
56
57  # Task
58  Generate 10 new image feature functions in Python that:
59
60  1. Help us discriminate between different image classes, hopefully with samples before and
        after the optimal split point having the lowest possible variance between their
        classifications.
61  2. Return a float value given an image.
62  3. Handle edge cases gracefully - won't crash on unusual images
63
64  Your task is to generate diverse, creative features that are relevant to explain the
        classifications for the image samples shown above. Focus on features that would help
        distinguish between samples of different classes. These features will be used in this
        decision tree that will predict the classification of a given image sample. Think about
        new features that would help such a predictor in the particular cases above, trying to
        add information that the already existing features shown above are missing.
65
66  # Code Requirements
67  - Use single quotes for docstrings: "description here"
68  - No markdown code blocks
69  - No explanatory text after the function
70  - Each function should be complete and standalone, and return a float
71
72  # Output Format
73  Generate exactly 10 features in this format:
74
75  def feature(image: np.ndarray) -> float:
76      "Simple, clear description of what this feature measures"
77      # ... Calculate and return the feature value
78      return result
79
80  def feature(image: np.ndarray) -> float:
```

```
81      "Another feature description"
82      # ... Calculate and return the feature value
83      return result
84
85  The body of the function can be anything, but the first line (function declaration) should be
        identical to those examples above, and the second line should be a one-line docstring.
        Don't output explanatory text - just the function definitions as shown above.
86
87  # Current decision tree node
88  You are currently focusing on features that explain the image classifications in the following
        subtree of a decision tree:
89
90  [root]
91   -> value > 0.344 for "Ratio of edge pixels to the total number of pixels"  -> value > 72.865
        for "Computes the brightness of the central region as a percentage of the whole image"
        -> value > 0.805 for "Determines the ratio of vertical to horizontal gradients in the
        image, indicating edge direction"
92
93  # Image samples
94  Here are examples of image samples in this subtree, along with their target classifications:
95
96  Sample (Target: 9)
97  Sample (Target: 6)
98  Sample (Target: 7)
99  Sample (Target: 4)
100 Sample (Target: 4)
101
102 Optimize for producing discriminant features that are novel compared to the existing features
        used to arrive at this subtree. Focus on explaining the differences between the image
        samples shown above.
```

### F.2.3 D-ID3 - EXAMPLE FEATURES

```
1   def feature(image: np.ndarray) -> float:
2       "Ratio of edge pixels to the total number of pixels"
3       if len(image.shape) == 3:
4           gray = np.mean(image, axis=2)
5       else:
6           gray = image
7       edges = np.gradient(gray.astype(float))
8       edge_pixels = np.count_nonzero(edges[0]) + np.count_nonzero(edges[1])
9       total_pixels = gray.size
10      return float(edge_pixels / total_pixels)
11
12
13  def feature(image: np.ndarray) -> float:
14      "Computes the brightness of the central region as a percentage of the whole image"
15      if len(image.shape) == 3:
16          gray = np.mean(image, axis=2)
17      else:
18          gray = image
19      h, w = gray.shape
20      center_region = gray[h//4:3*h//4, w//4:3*w//4]
21      total_brightness = np.sum(gray)
22      center_brightness = np.sum(center_region)
23      return float(center_brightness) / (total_brightness + 1e-7) * 100
24
25
26  def feature(image: np.ndarray) -> float:
27      "Determines the ratio of vertical to horizontal gradients in the image, indicating edge
         direction"
28      if len(image.shape) == 3:
29          gray = np.mean(image, axis=2)
30      else:
31          gray = image
32      grad_y, grad_x = np.gradient(gray)
33      vertical_grad = np.sum(np.abs(grad_x))
34      horizontal_grad = np.sum(np.abs(grad_y))
35      return float(vertical_grad) / (horizontal_grad + 1e-7)  # Avoid division by zero
36
37  def feature(image: np.ndarray) -> float:
38      "Calculates the ratio of bright pixels (above a threshold) to total pixels in the image"
39      if len(image.shape) == 3:
40          gray = np.mean(image, axis=2)
41      else:
42          gray = image
43      threshold = 200  # A threshold for bright pixels
44      bright_pixels = np.count_nonzero(gray > threshold)
45      total_pixels = gray.size
46      return float(bright_pixels) / total_pixels
```

```
47
48 def feature(image: np.ndarray) -> float:
49     "Measures the contrast of the image based on the standard deviation of pixel intensities"
50     if len(image.shape) == 3:
51         gray = np.mean(image, axis=2)
52     else:
53         gray = image
54     return float(np.std(gray))
```

## F.3   TEXT CLASSIFICATION

### F.3.1   F2

```
1 You are an expert text analysis programmer creating evaluation features for a machine learning
      model that classifies text.
2
3
4 # Text Processing API Documentation
5
6 The features receive text as a string, so you can use any string methods and text processing
      functions.
7
8 ## String Methods
9 - text.lower(), text.upper(): Case conversion
10 - text.strip(): Remove whitespace
11 - text.split(delimiter): Split into list
12 - text.count(substring): Count occurrences
13 - text.startswith(prefix), text.endswith(suffix): Check prefixes/suffixes
14 - text.find(substring): Find position of substring
15 - text.replace(old, new): Replace text
16
17 ## Text Analysis
18 - len(text): Length of text
19 - text.isdigit(), text.isalpha(), text.isalnum(): Character type checks
20 - sum(1 for c in text if c.isupper()): Count uppercase letters
21 - text.split(): Split on whitespace to get words
22
23 ## Regular Expressions (re module)
24 - re.findall(pattern, text): Find all matches
25 - re.search(pattern, text): Find first match
26 - re.sub(pattern, replacement, text): Replace patterns
27 - len(re.findall(r'\w+', text)): Count words
28 - len(re.findall(r'[.!?]', text)): Count sentences
29
30 ## Useful Patterns
31 - Word count: len(text.split())
32 - Sentence count: text.count('.') + text.count('!') + text.count('?')
33 - Average word length: sum(len(word) for word in text.split()) / len(text.split())
34 - Punctuation density: sum(1 for c in text if not c.isalnum() and not c.isspace()) / len(text)
35
36 ## Example Feature Function
37 def feature(text: str) -> float:
38     "Average word length in the text"
39     words = text.split()
40     if not words:
41         return 0.0
42     return sum(len(word) for word in words) / len(words)
43
44
45 ## Current Feature Database
46 Here are some existing features and their performance (Performance improvement = benefit from
      that feature, higher is better):
47
48 Feature:
49 def feature(text: str) -> float:
50     'Average number of words per sentence (sentences split on . ! ?)'
51     import re
52     if not text or not text.strip():
53         return float(0.0)
54     # Split on one or more sentence-ending punctuation and filter empties
55     sentences = [s.strip() for s in re.split(r'[.!?]+', text) if s.strip()]
56     if not sentences:
57         return float(0.0)
58     total_words = sum(len(s.split()) for s in sentences)
59     return float(total_words / len(sentences))
60
61
62 Importance: 0.057
63 ---
```

```
64
65
66  Feature:
67  def feature(text: str) -> float:
68      'Average character-uniqueness per word (unique chars / word length), averaged over words'
69      words = [w for w in text.split() if any(ch.isalnum() for ch in w)]
70      if not words:
71          return float(0.0)
72      ratios = []
73      for w in words:
74          chars = [c for c in w if not c.isspace()]
75          if not chars:
76              continue
77          unique = len(set(chars))
78          ratios.append(unique / len(chars))
79      if not ratios:
80          return float(0.0)
81      return float(sum(ratios) / len(ratios))
82
83
84  Importance: 0.057
85  ---
86
87
88  Feature:
89  def feature(text: str) -> float:
90      'Proportion of alphabetic characters that are uppercase'
91      if not text:
92          return float(0.0)
93      alpha_chars = [c for c in text if c.isalpha()]
94      if not alpha_chars:
95          return float(0.0)
96      upper_count = sum(1 for c in alpha_chars if c.isupper())
97      return float(upper_count / len(alpha_chars))
98
99
100 Importance: 0.083
101 ---
102
103
104 Feature:
105 def feature(text: str) -> float:
106     'Proportion of long words (length > 7) among all words'
107     words = [w for w in text.split() if w]
108     if not words:
109         return float(0.0)
110     long_count = sum(1 for w in words if len(w) > 7)
111     return float(long_count / len(words))
112
113
114 Importance: 0.194
115 ---
116
117
118 Feature:
119 def feature(text: str) -> float:
120     'Punctuation characters per word (punctuation = not alnum and not whitespace)'
121     if not text or not text.strip():
122         return float(0.0)
123     words = text.split()
124     if not words:
125         return float(0.0)
126     punct_count = sum(1 for c in text if not c.isalnum() and not c.isspace())
127     return float(punct_count / len(words))
128
129
130 Importance: 0.096
131 ---
132
133
134 Feature:
135 def feature(text: str) -> float:
136     'Proportion of sentences that are questions (based on ? count over total sentence
         terminators)'
137     if not text or not text.strip():
138         return float(0.0)
139     question_marks = text.count('?')
140     sentence_terminators = text.count('.') + text.count('!') + text.count('?')
141     if sentence_terminators == 0:
142         return float(0.0)
143     return float(question_marks / sentence_terminators)
```

```
144
145  Importance: 0.023
146  ---
147
148
149  Feature:
150  def feature(text: str) -> float:
151      'Type-token ratio: unique word tokens / total words (case-insensitive, alphanumeric tokens
        )'
152      import re
153      tokens = re.findall(r'\w+', text.lower())
154      if not tokens:
155          return float(0.0)
156      unique = len(set(tokens))
157      return float(unique / len(tokens))
158
159
160  Importance: 0.089
161  ---
162
163
164  Feature:
165  def feature(text: str) -> float:
166      'Ratio of tokens that contain at least one digit'
167      tokens = text.split()
168      if not tokens:
169          return float(0.0)
170      num_with_digit = sum(1 for t in tokens if any(ch.isdigit() for ch in t))
171      return float(num_with_digit / len(tokens))
172
173
174  Importance: 0.165
175  ---
176
177
178  Feature:
179  def feature(text: str) -> float:
180      'Longest run of the same character normalized by text length'
181      if not text:
182          return float(0.0)
183      max_run = 1
184      current_run = 1
185      prev = text[0]
186      for c in text[1:]:
187          if c == prev:
188              current_run += 1
189              if current_run > max_run:
190                  max_run = current_run
191          else:
192              current_run = 1
193              prev = c
194      return float(max_run / max(1, len(text)))
195
196
197  Importance: 0.126
198  ---
199
200
201  Feature:
202  def feature(text: str) -> float:
203      'Stopword density: fraction of tokens that are common English stopwords'
204      stopwords = {
205          'the','and','is','in','it','of','to','a','an','that','this','for','on','with',
206          'as','by','at','from','or','be','are','was','were','has','have','not','but',
207          'they','their','you','I'
208      }
209      tokens = [t.lower().strip(".,!?;:\"'()[]") for t in text.split()]
210      if not tokens:
211          return float(0.0)
212      stop_count = sum(1 for t in tokens if t and t in stopwords)
213      return float(stop_count / len(tokens))
214
215
216  Importance: 0.111
217  ---
218
219
220  ## Task
221  Generate 10 NEW text features that:
222
223  1. Are different from existing features
```

```
224  2. Capture useful textual patterns
225  3. Return float values
226  4. Handle edge cases gracefully - Won't crash on unusual texts
227  5. Use simple, short docstrings - Use single quotes, not triple quotes
228  6. Are efficient to compute
229
230  Your task is to generate diverse, creative features that capture different aspects of text
         content for classification. Focus on features that would help distinguish between
         different text classes. These features will be used as input to a learned model that
         predicts target values from text.
231
232  ## IMPORTANT CODE REQUIREMENTS
233  - Use SINGLE quotes for docstrings: "description here"
234  - NO triple quotes (""") anywhere in the code
235  - NO markdown code blocks
236  - NO explanatory text after the function
237  - Each function should be complete and standalone
238
239  ## Output Format
240  Generate exactly 10 features in this format:
241
242  def feature(text: str) -> float:
243      "Clear description of what this feature measures"
244      # ... Calculate and return the feature value
245      return float(result)
246
247  def feature(text: str) -> float:
248      "Another feature description"
249      # ... Calculate and return the feature value
250      return float(result)
251
252  The body of the functions can be anything, but the first line (function declaration) should be
         identical to those examples above (always 'def feature(...)'), and the second line
         should be a one-line docstring. Don't output explanatory text - just the function
         definitions as shown above.
```

### F.3.2   D-ID3 - PROMPT

```
1   You are an expert text analysis programmer creating feature functions to help a machine
        learning model perform text classification.
2
3   This is a classification task with the following classes: 0: human-written text, 1: AI-
        generated text.
4
5   Your task is to write a feature function that helps discriminate between the text samples
        given below.
6   A feature function is a Python function that takes a text string and computes a feature out of
         the text. It should return a float, but note that a feature could also be effectively
         boolean-valued (0.0 or 1.0), or integer-valued, even if its type is float.
7
8   You have access to the following API from text processing libraries:
9
10
11  # Text Processing API Documentation
12
13  The features receive text as a string, so you can use any string methods and text processing
        functions.
14
15  ## String Methods
16  - text.lower(), text.upper(): Case conversion
17  - text.strip(): Remove whitespace
18  - text.split(delimiter): Split into list
19  - text.count(substring): Count occurrences
20  - text.startswith(prefix), text.endswith(suffix): Check prefixes/suffixes
21  - text.find(substring): Find position of substring
22  - text.replace(old, new): Replace text
23
24  ## Text Analysis
25  - len(text): Length of text
26  - text.isdigit(), text.isalpha(), text.isalnum(): Character type checks
27  - sum(1 for c in text if c.isupper()): Count uppercase letters
28  - text.split(): Split on whitespace to get words
29
30  ## Regular Expressions (re module)
31  - re.findall(pattern, text): Find all matches
32  - re.search(pattern, text): Find first match
33  - re.sub(pattern, replacement, text): Replace patterns
34  - len(re.findall(r'\w+', text)): Count words
35  - len(re.findall(r'[.!?]', text)): Count sentences
36
```

```
37 ## Useful Patterns
38 - Word count: len(text.split())
39 - Sentence count: text.count('.') + text.count('!') + text.count('?')
40 - Average word length: sum(len(word) for word in text.split()) / len(text.split())
41 - Punctuation density: sum(1 for c in text if not c.isalnum() and not c.isspace()) / len(text)
42
43 ## Example Feature Function
44 def feature(text: str) -> float:
45     "Average word length in the text"
46     words = text.split()
47     if not words:
48         return 0.0
49     return sum(len(word) for word in words) / len(words)
50
51
52 # Task
53 Generate 10 new text feature functions in Python that:
54
55 1. Help us discriminate between different text classes, hopefully with samples before and
       after the optimal split point having the lowest possible variance between their
       classifications.
56 2. Return a float value given a text string.
57 3. Handle edge cases gracefully - won't crash on unusual texts
58
59 Your task is to generate diverse, creative features that are relevant to explain the
       classifications for the text samples shown above. Focus on features that would help
       distinguish between samples of different classes. These features will be used in this
       decision tree that will predict the classification of a given text sample. Think about
       new features that would help such a predictor in the particular cases above, trying to
       add information that the already existing features shown above are missing.
60
61 # Code Requirements
62 - Use single quotes for docstrings: "description here"
63 - No markdown code blocks
64 - No explanatory text after the function
65 - Each function should be complete and standalone, and return a float
66
67 # Output Format
68 Generate exactly 10 features in this format:
69
70 def feature(text: str) -> float:
71     "Simple, clear description of what this feature measures"
72     # ... Calculate and return the feature value
73     return result
74
75 def feature(text: str) -> float:
76     "Another feature description"
77     # ... Calculate and return the feature value
78     return result
79
80 The body of the function can be anything, but the first line (function declaration) should be
       identical to those examples above, and the second line should be a one-line docstring.
       Don't output explanatory text - just the function definitions as shown above.
81
82 # Current decision tree node
83 You are currently focusing on features that explain the text classifications in the following
       subtree of a decision tree:
84
85 [root]
86  -> value > 4.468 for "Average character length of words in the text"  -> value > 0.024 for "
       Calculates the proportion of text that is in passive voice"  -> value < 26.000 for "
       Assesses the use of passive voice constructions in the text"
87
88 # Text samples
89 Here are examples of text samples in this subtree, along with their target classifications:
90
91 Sample: 'The Strange Tank
92
93 John awoke with a start. He was submerged in a tank of pink, viscous liquid. He thrashed
       around in a panic, trying to determine which way was up. He finally surfaced, gasping for
       ai...', (Target: 1)
94 Sample: 'The old map crumbled at the edges as Juan carefully unrolled it across the table. His
       grandfather had given him the map many years ago, telling him it showed the true history
       of this land. A history m...', (Target: 1)
95 Sample: 'The call came at midnight, just as I finished cleaning and assembling my sniper rifle
       . My burner phone buzzed, the one dedicated only to receiving instructions. A robotic
       female voice spoke: "Contract...', (Target: 1)
96 Sample: '"Bang Bang Baby" can be labeled a science fiction musical as it incorporates elements
       of both genres in an exciting plot. After watching the trailer first, I was surprised by
       the energetic nature of t...', (Target: 0)
```

```
97  Sample: 'The red dust swirled around John as he took his first step onto the rust-colored soil
            of Mars. After decades of training and a perilous 9 month journey through the inky black
            void of space, he had fin...', (Target: 1)
98
99  Optimize for producing discriminant features that are novel compared to the existing features
            used to arrive at this subtree. Focus on explaining the differences between the text
            samples shown above.
```

### F.3.3 D-ID3 - EXAMPLE FEATURES

```python
 1  def feature(text: str) -> float:
 2      "Average character length of words in the text"
 3      words = text.split()
 4      if not words:
 5          return 0.0
 6      return sum(len(word) for word in words) / len(words)
 7
 8  def feature(text: str) -> float:
 9      "Calculates the proportion of text that is in passive voice"
10      passive_pattern = r'\b(?:is|was|were|be|being|been) \w+\b'
11      passive_count = len(re.findall(passive_pattern, text))
12      return float(passive_count) / len(text.split()) if text.split() else 0.0
13
14  def feature(text: str) -> float:
15      "Assesses the use of passive voice constructions in the text"
16      passive_voice = re.findall(r'\b(is|are|was|were|be|being|been)\s+\w+\b', text)
17      return float(len(passive_voice))
18
19  def feature(text: str) -> float:
20      "Counts the number of transition words to evaluate the flow of text"
21      transition_words = set(['however', 'therefore', 'moreover', 'furthermore', 'nevertheless',
            'consequently'])
22      words = text.lower().split()
23      count = sum(1 for word in words if word in transition_words)
24      return float(count)
25
26  def feature(text: str) -> float:
27      "Measures the proportion of first-person pronouns in the text"
28      first_person_pronouns = set(['I', 'me', 'my', 'mine', 'we', 'us', 'our', 'ours'])
29      words = text.lower().split()
30      count = sum(1 for word in words if word in first_person_pronouns)
31      return float(count) / len(words) if words else 0.0
```

