# OpenReview forum: "Programmatic Representation Learning with Language Models"
_ICLR.cc/2026/Conference — Submitted to ICLR 2026_

### Official Review · Reviewer_kY8q · 2025-10-26

**Soundness:** 2
**Presentation:** 2
**Contribution:** 2
**Rating:** 2
**Confidence:** 3

**Summary:**

This paper introduces LeaPR (Learned Programmatic Representations) models, which stack arbitrary features represented as code and decision tree predictors. The authors show that their proposed method is competitive with neural networks (without using neural models) on a few set of benchmark.

**Strengths:**

1. Using Large Language Models for feature generation is an important research directory.

2. Using some evolutionary algorithms like FunSearch to such a certain application is quite interesting.

**Weaknesses:**

1. My biggest concern is that I think authors have to deeply do research on feature generation works. Generating raw features to enhance a prediction model (both for non-neural models and neural models) are widely proposed. For example, OpenFE and AutoFeat use classical learning algorithms to generate raw features. Moreover, CAAFE and OCTree proposes to use LLMs for feature generation. Specifically, OCTree also uses an evolutionary algorithm similar to FunSearch and AlphaEvolve. I suggest authors to carefully look at these works and clarify the difference of LeaPR compared to them. Also, it would be great to be involved as a baseline.

2. Baselines are too weak. Why did the authors only use Transformers as a baseline? There are much more neural-based models powerful than Transformers. For example, for the tabular dataset, TabPFN is known to be a SOTA model.

3. Is the used benchmark useful to show the interpretability of the model? I ask this because, the authors argue that their proposed models show better interpretability than neural-based models. What experiments support this statement? Or is this because LeaPR only uses tree-based models?

4. Number of benchmark is too low and outdated.

**Questions:**

See the above Weaknesses.

---

> ### Author Response · Authors · 2025-11-21
>
> We thank the reviewer for the comments! We address each of the concerns below.
>
> > My biggest concern is that I think authors have to deeply do research on feature generation works. Generating raw features to enhance a prediction model (both for non-neural models and neural models) are widely proposed. For example, OpenFE and AutoFeat use classical learning algorithms to generate raw features. Moreover, CAAFE and OCTree proposes to use LLMs for feature generation. Specifically, OCTree also uses an evolutionary algorithm similar to FunSearch and AlphaEvolve. I suggest authors to carefully look at these works and clarify the difference of LeaPR compared to them. Also, it would be great to be involved as a baseline.
>
> Thank you for highlighting these important related works! We’ve added detailed discussion of all of these methods to Appendix E and a reference in the main text (Section 2). In particular, the main distinction is that these are meant to work with **tabular datasets**, unlike our domains (text, images, chess, and now raw audio added in Appendix C).
>
> To clarify the methodological distinctions: OpenFE and AutoFeat use predefined operators to transform existing tabular columns, while CAAFE and OCTree use LLMs to create transformations of existing features in tabular datasets. All of these methods generate feature transformations for tabular data with pre-existing features, while LeaPR is more general, and generates full features (not feature transformations) directly from raw inputs. We did not include these methods as baselines since our domains contain unstructured inputs, with no existing named columns to transform, unlike in the tabular setting.
>
> > Baselines are too weak. Why did the authors only use Transformers as a baseline? There are much more neural-based models powerful than Transformers. For example, for the tabular dataset, TabPFN is known to be a SOTA model.
>
> We did not exclusively use only Transformers, since in our experiments in vision we compared against standard CNN baselines. However, for both text classification and chess, Transformers are the strongest models on those tasks, making that the natural baseline. For Ghostbuster (the text classification task), we reported all of the state-of-the-art methods, whereas for chess we compared against the recently proposed Transformer baseline because it was shown to outperform other models, such as the CNN used in AlphaZero.
>
> Regarding TabPFN, we believe there was a misunderstanding, since **we did not use any tabular dataset**.
>
> TabPFN is designed for tabular classification problems with pre-existing features: they train a transformer to perform in-context learning on synthetic tabular datasets, enabling predictions on **small** datasets (up to 1000 samples and up to 100 features). While TabPFN achieves strong performance within these constraints, it represents a fundamentally different approach: it is an end-to-end black-box neural network, whereas LeaPR generates programmatic features that can be directly inspected and validated. Additionally, TabPFN is limited to small datasets because it feeds the entire dataset as a sequence to the transformer, whereas LeaPR scales to datasets with **hundreds of thousands** of samples. For these reasons, we could not include it as a baseline in our experiments.
>
> As suggested by reviewers, we have added several other baselines, including two baselines using GPT-5.1 (a much stronger model than the LLMs we used with LeaPR), as well as the raw inputs with random forests when possible. Please check the updates in Section 4, Tables 1, 2 and 3.
>
> > Is the used benchmark useful to show the interpretability of the model? I ask this because, the authors argue that their proposed models show better interpretability than neural-based models. What experiments support this statement? Or is this because LeaPR only uses tree-based models?
>
> The interpretability comes not only from the tree-based model, but even more importantly from the features being represented as code, and we have found them generally very easy to understand. The interpretability aspect is best demonstrated in our case studies in Section 4.4, where we look at particular features driving predictions in text and image classification, and find that they can be easily analyzed.
>
> > Number of benchmark is too low and outdated.
>
> We note that each of our benchmarks showcases our model on a different input modality – we have also added a new evaluation on audio classification (see Appendix C.1, where LeaPR features outperform expert-designed features for this task). Moreover, the Ghostbuster dataset, where we match the neural state of the art, was published in NAACL 2024. The chess baseline we compared against was similarly published at NeurIPS 2024. Thus, these are not "outdated" benchmarks.
>
> We hope to have addressed your concerns regarding our paper. If not, please let us know, and we'd be happy to keep engaging in the discussion!

---

### Official Review · Reviewer_iR4d · 2025-10-28

**Soundness:** 2
**Presentation:** 2
**Contribution:** 3
**Rating:** 4
**Confidence:** 3

**Summary:**

This paper presents a method for obtaining interpretable classifiers by using language models to propose a set of feature extractors in the form of Python programs. The feature set is evolved iteratively by training a decision tree on top of the current feature set, and prompting the LLM to propose features to improve the decision tree accuracy. An extension of the method uses an ID3-style algorithm to find features that iteratively split nodes in the decision tree with high classification error. The method is evaluated on three domains: classifying chess positions, images, and text.

**Strengths:**

- I think this paper presents an interesting solution to the important problem of developing more interpretable machine learning models. Representing features as programs makes it possible to interpret features, edit the features after the fact, and take advantage of the strengths of state-of-the-art (but uninterpretable) language models.

- The method builds on prior work (FunSearch) but represents a meaningful extension: FunSearch searches for a single program, while this method has to search for a set of programs, representing different features, which is a non-trivial extension.

- The paper conducts experiments on several different domains (chess, images, and text), which demonstrates the generality of the approach. The performance is fairly impressive, with LeaPR performing more or less on par with neural network baselines.

- The paper presents a number of interesting examples of programs. These examples show that the method is useful for discovering interesting properties of training distributions---for example, that LM-generated text is more likely to contain a combination of ascii-quotation marks and curly quotation marks. I also appreciated the case study of debugging a spurious feature in the Waterbird dataset.

- I can see this method providing the basis for future work--for example, to learn hierarchical features, or simply to improve the prompting scheme for generating non-hierarchical feature extractor programs.

**Weaknesses:**

- The paper is missing some details about the formal definition of the methods. Figure 2 presents definitions of the two algorithms, but the subroutines in these algorithms are not formally defined (RandomKFeatures, ProposeFeatures, SplitError). Much of the exposition of the algorithms is presented informally in the text. This makes it difficult to understand exactly how the algorithms work, and to reason about the different design choices.


- The method is evaluated on very simple problems, where it is possible to obtain good performance with a small number of simple features (e.g. detecting LLM-generated text). It is not clear how well it will scale up to more complex domains. It would have been helpful to see results on a more challenging domain--for example, natural language inference, for text [1]. I don't think the method has to perform well on these datasets to be worthy of publication, but I think showing results on such datasets would help to illustrate the limitations of the method which could be improved in future work.


- It is not entirely clear to me why decision trees in particular are used. In my opinion, the paper would be stronger if the method was stated more generally, as a method for iteratively evolving (1) a set of features, and (2) a predictor that predicts outcomes based on the features. This would then make it possible to compare different choices of predictor, such as linear models, in addition to decision trees.


- The paper would benefit from more experiments illustrating the computational cost of this approach. For example, LLM-based program synthesis methods are known to benefit from a very high number of samples (see e.g. [2]). Some details about the number of iterations are reported in the appendix, but it would have been useful to see some empirical results about the relationship between number of samples and final performance.


- The paper argues that one of the advantages of this method is that it is less data intensive than neural networks. But there is no empirical experiment comparing the neural networks and LeaPR with different data budgets. One could also argue that one of the _benefits_ of neural networks is that they can take advantage of additional training data--if LeaPR does not improve with more data, this is arguable a limitation of the approach. This point is alluded to at the end of section 4.1, but it would be helpful to see some empirical analysis.


- In the image domain, LeaPR is trained without access to the underlying image data. The authors acknowledge this limitation in section 4.2, but it could be highlighted more prominently in the introduction.


Minor comments

- The terms "low-level" and "high-level" (as in "low-level inputs") in the introduction are not clearly defined.


Overall, I think this paper presents a promising idea and represents an exciting direction for future work. However, I think the paper needs some improvements before I could recommend accepting it--especially, to supply the missing formal definitions of the algorithms mentioned above. I would be happy to increase my score if these weaknesses could be addressed.



[1] Williams et al., 2018. A Broad-Coverage Challenge Corpus for Sentence Understanding through Inference.


[2] https://blog.redwoodresearch.org/p/getting-50-sota-on-arc-agi-with-gpt

**Questions:**

- Is there any reason that the feature programs are all given the same name (`def feature(...) -> float:`)?

- Modern frontier LLMs support image inputs. Did you try running the image experiments with training images included in the prompt?


- I generally found the related work section to be a good overview of the area, but it could also include related work on iterative library learning, such as DreamCoder [1], and some discussion of what makes a predictor interpretable (e.g. [2]).


[1] Ellis et al., 2023. DreamCoder: growing generalizable, interpretable knowledge with wake–sleep Bayesian program learning.

[2] Lipton, 2018. The mythos of model interpretability: In machine learning, the concept of interpretability is both important and slippery.

---

> ### Author Response · Authors · 2025-11-21
>
> We sincerely thank the reviewer for the several constructive comments and suggestions, and for the remark that our paper provides "exciting direction for future work"! We respond to each of the points inline below.
>
> > The paper is missing some details about the formal definition of the methods. Figure 2 presents definitions of the two algorithms, but the subroutines in these algorithms are not formally defined (RandomKFeatures, ProposeFeatures, SplitError). [...]
>
> Thank you for pointing this out! We have expanded on the definition of all of those subroutines to Appendix A.1. Please let us know if there are any details still unclear.
>
> > The method is evaluated on very simple problems, where it is possible to obtain good performance with a small number of simple features (e.g. detecting LLM-generated text). It is not clear how well it will scale up to more complex domains. It would have been helpful to see results on a more challenging domain--for example, natural language inference, for text [1]. I don't think the method has to perform well on these datasets to be worthy of publication, but [...] would help to illustrate the limitations of the method which could be improved in future work.
>
> Evaluating on more domains is always insightful, though we argue that the task of detecting LLM-generated text is not necessarily very simple. The state-of-the-art method on Ghostbuster, which we match with LeaPR, involves running the input text through an ensemble of large language models and training a classifier that uses their activations. Other recent methods are also very complex, such as GPT-Zero, which uses the curvature of the Hessian of a given LLM, with other baselines being fine-tuning a prediction head on top of existing models, and so on. The fact that LeaPR does so well on this task is not to be taken trivially - given the complexity of recently published baselines on this task, we don't think the natural expectation would be that a decision tree trained on interpretable features discovered fully automatically could do so well. Similarly, for chess, it is not possible to "obtain good performance with a small number of simple features".
>
> That said, running on SNLI is still informative and illustrates a particular current limitation of our approach. For text, since we did not allow programs to use special libraries, LeaPR currently struggles to encode highly semantic features (e.g., reliably detect synonyms). We evaluated D-ID3 with gpt-5-mini on SNLI, and achieved a test accuracy of 66.3% - for what the SNLI leaderboard denotes as "feature-based models", this is better than a feature-based model using only unlexicalized features (50.4%), but underperforms using both those and lexicalized features (78.2%). While LeaPR might improve from using  NLP libraries that would allow it to reason semantically about the text (e.g., having access to wordnet via NLTK, or spaCy), we reported the current result in Appendix C.2 and added a reference in the main text.
>
> > It is not entirely clear to me why decision trees in particular are used. In my opinion, the paper would be stronger if the method was stated more generally, as a method for iteratively evolving (1) a set of features, and (2) a predictor that predicts outcomes based on the features.
>
> We agree - the method is general and could be combined with any of the many available classical predictors (including SVMs, kNN models, besides linear models, etc). This is a very large space of possibilities, and we leave a broader exploration of these other model classes (with their potential advantages for particular tasks) for future work.
>
> > The paper would benefit from more experiments illustrating the computational cost of this approach. [...] it would have been useful to see some empirical results about the relationship between number of samples and final performance.
>
> This is a good suggestion, in line with reviewer KrKR. We have added a new analysis to the main paper that directly relates the number of LeaPR features (selected in 3 ways: the order in which D-ID3 proposes them, the longest, and the shortest in lines of code) and the final performance in each domain. See Section 4.4 for the full analysis. In summary, we find that (1) D-ID3 tends to find the most impactful features earlier, but (2) performance keeps steadily improving with more features, suggesting our method can keep finding features that improve upon its current predictor. Those trends hold across domains, whereas the relationship between feature complexity and performance is more nuanced, with more complex features tending to help more in some domains (e.g., image classification) and simpler features outperforming in others (e.g. chess).

---

> ### Author Response · Authors · 2025-11-21
>
> > The paper argues that one of the advantages of this method is that it is less data intensive than neural networks. But there is no empirical experiment comparing the neural networks and LeaPR with different data budgets. One could also argue that one of the benefits of neural networks is that they can take advantage of additional training data--if LeaPR does not improve with more data, this is arguable a limitation of the approach. This point is alluded to at the end of section 4.1, but it would be helpful to see some empirical analysis.
>
> Thank you for your suggestion. While we have not performed a comparison of LeaPR versus neural networks across varying training set sizes in all domains, our analysis in chess shows that a Transformer trained on 200x more data still underperforms LeaPR on that domain. Moreover, the new analysis we added in Section 4.4 shows that LeaPR’s performance improves steadily as more features are added to the predictor, and we don't seem to have saturated performance even at our scale.
>
> > In the image domain, LeaPR is trained without access to the underlying image data. The authors acknowledge this limitation in section 4.2, but it could be highlighted more prominently in the introduction.
>
> We agree, and have noted in the introduction, when we mention few-shot examples, that this only currently applies for domains where inputs that can be encoded as text.
>
> > The terms "low-level" and "high-level" (as in "low-level inputs") in the introduction are not clearly defined.
>
> We agree that these terms were unclear. We rephrased both the introduction (where we simply refer to images and text as unstructured inputs, not low-level), and the use of this term in the method description (we simply explain that individual features in chess are not informative enough for decision trees to make meaningful predictions, which is what we meant originally by "low-level"). Thank you for helping us clarify the text!
>
> > Is there any reason that the feature programs are all given the same name (def feature(...) -> float:)?
>
> This was simply a convenience for us to extract the feature from the model's response, and then manipulate it on our end.
>
> > Modern frontier LLMs support image inputs. Did you try running the image experiments with training images included in the prompt?
>
> We did not, due to the expensive cost of our prompts including images (over 40x the price with text only).
>
> > I generally found the related work section to be a good overview of the area, but it could also include related work on iterative library learning, such as DreamCoder [1], and some discussion of what makes a predictor interpretable (e.g. [2]).
>
> Thank you for the suggestion! We have updated our related work (section 2) to expand on both of these points, discussing the relationship to DreamCoder and the point on interpretability.
>
> > Overall, I think this paper presents a promising idea and represents an exciting direction for future work. However, I think the paper needs some improvements before I could recommend accepting it--especially, to supply the missing formal definitions of the algorithms mentioned above. I would be happy to increase my score if these weaknesses could be addressed.
>
> Thank you again for the constructive comments! We hope we managed to address your concerns with our updates. Please let us know If there's anything else we can provide, and we'd be happy to engage further.

---

> > ### Comment · Reviewer_iR4d · 2025-11-26
> >
> > Thank you for the detailed response. I think the additional methodological details and results will strengthen the paper (experiments on SNLI, analysis of performance vs. number of features in section 4.4). I have a few remaining comments:
> >
> > > Definition of all of those subroutines to Appendix A.1
> >
> > Thank you for adding these. I think some of these definitions could still be stated more formally--especially ProposeFeatures. For example, could you present the templates used for generating the "context" in the two different cases? (It's difficult to read the full prompts in appendix F.)
> >
> > > Results on SNLI
> >
> > Thank you for adding this experiment and discussion. I would like to note that an accuracy of 66.3 is very low--according to [1], you can get this accuracy with a FastText classifier trained only on the hypothesis (without seeing the premise). I agree this could be due in part to the method not having access to semantic embeddings, but [2] show that you can also get very high accuracy (on MNLI) using simple heuristics, like a word overlap heuristic, which should be possible to express with this approach. So it's possible that the low accuracy is also related to other aspects of the method, like prompt design or different approaches to feature splitting.
> >
> > [1] Gururangan et al., 2018. Annotation Artifacts in Natural Language Inference Data. https://aclanthology.org/N18-2017/.
> >
> > [2] McCoy et al., 2019. Right for the Wrong Reasons: Diagnosing Syntactic Heuristics in Natural Language Inference. https://arxiv.org/abs/1902.01007.
> >
> > Given the additional results and methodological details, I will increase my score from 4 to 6. I still think the presentation of the paper could be improved--especially, to present the method more generally, to make it easier to reason about ablating or redesigning different components in the algorithm. I think the paper would also benefit from addressing the comments from the other reviewers, especially about better contextualizing the method with respect to other work on feature learning.

---

### Official Review · Reviewer_KrKR · 2025-10-28

**Soundness:** 2
**Presentation:** 3
**Contribution:** 2
**Rating:** 2
**Confidence:** 5

**Summary:**

The authors present LeaPR, a hypothesis class that combines programmatically generated features (represented as code) with tree-based predictors. The features are synthesized by Large Language Models (LLMs), leveraging their ability to write interpretable, domain-specific code. The paper introduces two algorithms for learning LeaPR models: (1) F2, an adaptation of existing feature-search method FunSearch, where LLMs generate global features based on feature importance, optimized for ensemble predictors such as Random Forests; and (2) D-ID3, a novel algorithm inspired by ID3 decision tree learning, where the model dynamically requests new LLM-generated features during tree construction to improve local splits.

The authors evaluate LeaPR on three domains, including chess position evaluation, image classification, and text classification. They show that the resulting models (sometimes) achieve comparable predictive performance compared to neural networks while retaining interpretable features. The paper also includes snippets of the learned code-based features and the prompts used for feature generation.

**Strengths:**

1. Leveraging LLMs to generate interpretable, programmatic features for classical models (like Trees) is an appealing approach that combines symbolic interpretability with LLM reasoning capabilities. Also, features represented as code snippets are human-readable and can be inspected or audited, improving model transparency. Authors present the D-ID3 algorithm, which incorporates feature generation dynamically during training, a creative extension of decision tree learning.

2. Authors present experiments on diverse tasks (text, images, chess) show that LeaPR can be used across modalities. They also share the LLM prompts and generated features that enhance reproducibility and interpretability.

**Weaknesses:**

1. Limited novelty: The general idea of using LLMs to generate features, either as code (Python functions) or as text/natural-language, has been explored in prior work since 2023 [1, 2, 3]. Several existing methods already use LLMs to produce interpretable or programmatic features for succinct predictors such as decision trees or linear models. Therefore, while D-ID3 provides an additional algorithmic contribution, the overall framework is not fully novel, and must be compared against existing frameworks.

[1] Singh, C., Morris, J., Rush, A. M., Gao, J., & Deng, Y. (2023, December). Tree prompting: Efficient task adaptation without fine-tuning. In Proceedings of the 2023 Conference on Empirical Methods in Natural Language Processing (pp. 6253-6267).
[2] Khandelwal, A., Pavlick, E., & Sun, C. (2023). Analyzing modular approaches for visual question decomposition. arXiv preprint arXiv:2311.06411.
[3] Chan, K. H. R., Chattopadhyay, A., Haeffele, B. D., & Vidal, R. (2023). Variational Information Pursuit with Large Language and Multimodal Models for Interpretable Predictions. arXiv preprint arXiv:2308.12562.

2. Missing baselines: The paper does not compare LeaPR against existing LLM-based feature generation methods, both code-based and text-based, which weakens claims of originality and performance. There is no comparison against LLMs performing direct reasoning or chain-of-thought (CoT) inference, which are strong baselines for structured prediction and may already capture interpretable intermediate reasoning steps without explicit feature synthesis. The study omits traditional feature-engineering baselines with hand-crafted expert features and does not examine whether LLM-generated features complement or improve upon human-designed ones. For predictive performance, the authors should include baselines using direct LLM inference (with the LLM feature-generators: GPT-4o mini and GPT-5 mini) and CoT reasoning to assess the true benefit of learned programmatic representations. For interpretability, results could also be compared against simpler interpretable models such as shallow decision trees or logistic regression with a small number of features.

3. Limited parameter exploration: There is limited analysis of how the number, complexity, or quality of generated features affects model performance and interpretability. The computational cost of using LLMs for feature synthesis is not discussed or compared against traditional feature extraction or neural representation learning.

4. Limited practical applicability: It is not straightforward to see how the chosen tasks could be applied in more practical settings.

**Questions:**

1. Can the LeaPR framework incorporate human-in-the-loop feedback to refine or prune LLM-generated features, thereby improving interpretability and reducing redundancy?
2. How does LeaPR handle potentially incorrect or low-quality features generated by the LLM, and are there mechanisms to automatically detect or discard unhelpful code-based features during training?

---

> ### Author Response · Authors · 2025-11-21
>
> We thank the reviewer for the suggestions and constructive remarks! We address each of the questions below.
>
> > Limited novelty: The general idea of using LLMs to generate features, either as code (Python functions) or as text/natural-language, has been explored in prior work since 2023 [1, 2, 3]. Several existing methods already use LLMs to produce interpretable or programmatic features for succinct predictors such as decision trees or linear models.
>
> We thank the reviewer for pointing these out! We incorporated these into our related work section (see our updated "Interpretability of Neural Networks" paragraph). We note, however, that **none of these works explore the idea of features as code**, as the reviewer implied. This is an important point that we emphasized now in our discussion of related work. In particular:
>
> * In TreePrompting [1], at inference time, the language model is prompted to answer a question at each decision node, and the data point is routed to lower nodes until it reaches a leaf. The LLM is not producing a feature, but rather an end-to-end decision, and there is *no code generation involved*. Note that, since the LLM is called at inference time, this requires the LLM to be able to receive the data point and answer the question. (e.g., this would not apply in a domain such as audio classification, which we have also evaluated on for the rebuttal – see results in Appendix C).
> * In ViperGPT and follow-up works [2], the LLM is queried at inference time to produce a specialized predictor, using small neural network modules as appropriate, to answer the question (e.g., their "How many black cats are in the image?" example in the Viper paper). But a different program has to be produced for each test data point. Again, *the program is a predictor, and not a feature*.
> * In FM+V-IP [3], the LLM generates questions in natural language at training time, and some neural module (CLIP, in their case) still has to answer the questions at inference time. This work *does not involve code generation*.
>
> In particular, in all of these papers, a neural network is still always involved during inference time. In sharp contrast, our methods use language models during training, but the final predictors (the LeaPR models, for which we give two training algorithms) are neural network-free. Our main claim to novelty is learning this new class of models, which is equipped with representation learning capabilities but does not require neural networks at inference time.
>
> > Missing baselines: [...]
>
> We have added several additional baselines, as suggested by you and other reviewers. In particular, we added two baselines based on the recently released GPT-5.1 model (much more performant that gpt-5-mini, which is what we used with LeaPR): Program of Thoughts, which also leverages code generation, as well as Google's original FunSearch, which we also used to evolve programs as predictors. We also added a baseline with the same random forest we used, but instead trained on simple/raw domain features (rather than our learned features). LeaPR with gpt-5-mini outperforms all of these baselines across all domains.
>
> Finally, to compare against a random forest baseline with expert-designed features, which were not available in the original domains we tested on, we ran an experiment on the audio classification dataset ESC-50. The dataset paper reports a random forest baseline trained on hand-designed audio features performing at 44.30% top-1 accuracy (note that this is a balanced 50-class classification problem, thus random performance would be 2%). We applied D-ID3 with gpt-5-mini on this dataset, and achieved an accuracy of 64.1%, significantly better than random forests paired with the hand-written expert features.

---

> > ### Author Response · Authors · 2025-11-21
> >
> > > Limited parameter exploration: There is limited analysis of how the number, complexity, or quality of generated features affects model performance and interpretability
> >
> > This is a good suggestion, in line with reviewer KrKR and iR4d. We have added a new analysis to the main paper that directly relates the number of LeaPR features (selected in 3 ways: the order in which d-id3 proposes them, the longest, and the shortest in lines of code) and the final performance in each domain. See Section 4.4 for the full analysis. In summary, we find that (1) D-ID3 tends to find the most impactful features earlier, but (2) performance keeps steadily improving with more features, suggesting our method can keep finding features that improve upon its current predictor. Those trends hold across domains, whereas the relationship between feature complexity and performance is more nuanced, with more complex features tending to help more in some domains (e.g., image classification) and simpler features outperforming in others (e.g. chess).
> >
> > Regarding computational cost, we provide more details in Appendix A.2, including LLM API costs per domain. Importantly, LeaPR’s inference cost is negligible compared to neural baselines–our models execute simple Python functions rather than large neural networks. For Chess, we compared LeaPR against a 270M-parameter transformer (Section 4.1, Table 1) that requires substantially more compute during both training and inference. The finding that D-ID3 discovers the most impactful features early also has practical implications for computational budget: users can terminate feature generation earlier (if computational budget is limited) and still obtain strong performance.
> >
> > > Limited practical applicability: It is not straightforward to see how the chosen tasks could be applied in more practical settings.
> >
> > Our tasks were meant to be representative of supervised learning -- a very general paradigm, and they cover both regression and classification, across 3 different modalities (4 with audio, which we added in Appendix C). Supervised learning has extensive practical applications, and we chose tasks to understand the generality of our method. However, we point that the particular task of text classification we demonstrated (detecting AI-generated text) is one with *highly practical application* nowadays: for instance, many production software platforms (e.g., for education) already ship with an AI-generated text classifier. In particular, practitioners have found the opaqueness of already deployed neural classifiers to be negative, since they encode biases that can go unnoticed before deployment (e.g. see https://www.cell.com/patterns/fulltext/S2666-3899(23)00130-7).
> >
> > > Can the LeaPR framework incorporate human-in-the-loop feedback to refine or prune LLM-generated features, thereby improving interpretability and reducing redundancy?
> >
> > Yes, this is certainly possible. Our case studies (see Section 4.5) were meant to demonstrate workflows where humans can readily inspect, understand current features and predictions, and "debug" the model. While we did not explore directly intervening in the model using the insights learned from these analyses, this is certainly an option that practitioners have with LeaPR models - both refining features, removing undesirable ones (e.g., spuriously correlated features), or using these observations to provide natural language feedback in the prompt. This is a very large space of possible interactions, and we leave this more through exploration for future work.
> >
> > > How does LeaPR handle potentially incorrect or low-quality features generated by the LLM, and are there mechanisms to automatically detect or discard unhelpful code-based features during training?
> >
> > We automatically filter out features that are either unstable (e.g., raise exceptions, return infinity/NaN for some corner cases, etc) or too slow, by testing them on the training set first - we run them in a separate process for sandboxing the training process. This is incorporated in step K in Algorithm 1 and L in Algorithm 2, and we added clarifications for those steps in Appendix A.1. If a feature is unhelpful, in the sense of simply not providing predictive signal, the decision tree trainer will simply not rely on it, which can be detected post-hoc by importance scores (e.g., SHAP, as we used in the case studies in Section 4.5).
> >
> > We hope we were able to address your concerns, and we'd be happy to engage further if you have any outstanding questions!

---

> > > ### Comment · Reviewer_KrKR · 2025-11-25
> > >
> > > I thank the reviewers for their thorough reply, in particular the positioning of the paper with respect to the studies I discussed. They also have added additional experiments with a newer GPT-5.1, and a new audio task which allowed them to compare their method with expert features. Nonetheless, they have not addressed my concern with respect to comparing their method with other similar method. For the reasons listed above, I'll raise my score to 4.

---

> > > > ### Author Response · Authors · 2025-11-25
> > > >
> > > > We sincerely thank the reviewer for taking our response into account, and for raising their score!
> > > >
> > > > We would be happy to provide a direct comparison to an existing method that would work out of the box with our datasets, if the reviewer is aware of such a method. However, to the best of our knowledge, our work is the first to propose generating LLM-synthesized features *for unstructured data* (e.g., text, or audio, or images). While a series of recent works (e.g., CAAFE, OCTree, both published at NeurIPS 2023 and 2024, respectively) can be applied to generate features in the *tabular case*, they make assumptions that are specific to this setting. We have added an extensive discussion of such methods in both our related work section and Appendix E. However, to keep the response self-contained, we here describe why CAAFE and OCTree, as two recent representatives of the work on feature engineering for tabular data, do not readily apply in our setting. Quotes below are from their respective papers.
> > > >
> > > > * **CAAFE (NeurIPS 2023)**: This method repeatedly prompts the LLM to generate new code that transforms existing features in the dataset, by providing it with:
> > > >   * "Feature names adding contextual information and allowing the LLM to generate code to
> > > > index features by their names". In contrast, our unstructured inputs do not start with named features. For instance, in text classification, the input is just a string. Our methods still work in this setting since they do not rely on feature names.
> > > >   * "Data types (e.g. float, int, category, string) - this adds information on how to handle a feature in the generated code". In our case, the data types are generally uninformative. For instance, in audio classification, all of the 220,500 input dimensions are of type "float".
> > > >   * "Percentage of missing values" -- this again only applies in the tabular setting. For instance, in our case, all images have all pixels in them.
> > > >   * "10 random rows from the dataset" -- this requires inputs to be given to the LLM in context. In contrast, our methods do not require this, and work in modalities (e.g., audio) that the LLM might not be able to receive as input, and nonetheless can write code to manipulate.
> > > >
> > > > Moreover, note that generally the information above is given *per input feature*: name, type, percentage of missing values for that feature. Our methods work in extremely high-dimensional settings (as showcased in audio), where giving feature-level information in context would be completely prohibitive.
> > > >
> > > > * **OCTree (NeurIPS 2024)**: similarly, OCTree makes several assumptions tailored to the tabular setting:
> > > >   * To start, its initial prompt, like CAAFE, also includes per-feature dataset statistics (see their Appendix A.1, referenced in the method description in Section 3.2), which, as we described above, is impossible in our setting with unstructured inputs.
> > > >   * In each iteration, "OCTree then generates an initial rule r0 for deriving a new column feature from the original set of columns C". As explained, in our setting, there is not necessarily a starting set of "columns" C.
> > > >   * To handle datasets where columns don't already have language descriptions (which is all of our settings), OCTree reduces this case to its default setting where columns have names by assigning "non-descriptive column names, such as C = {‘x1’, ‘x2’, . . . , ‘x5’} for a dataset with M = 5 columns.". Then, these names are used in the prompts for LLMs to generate simple rules on how to derive new columns. This again either does not apply or is extremely impractical in datasets that our methods work on: for instance, in audio, this would again entail generating over 200,000 of these "non-descriptive names" and including them in prompts.
> > > >
> > > > For these and other reasons, we cannot directly compare with these methods that were designed with tabular datasets in mind. These methods would require substantial adaptation, and then we would not be comparing with the original methods anymore, but rather with variants that would look more like F2 and D-ID3 than like OCTree or CAAFE.
> > > >
> > > > We hope this clarifies. If not, or if the reviewer had any other specific baseline in mind that would work in our cases, we will do our best to provide a fair comparison within the rebuttal period. We thank the reviewer again for the engagement!

---

### Official Review · Reviewer_idVN · 2025-11-01

**Soundness:** 2
**Presentation:** 3
**Contribution:** 3
**Rating:** 4
**Confidence:** 3

**Summary:**

This paper is best described as building of FunSearch [1] to introduce a binary tree version of this method

FunSearch proposes using LLMs in an unconventional way. It exploits the fact that LLMs can code very well to use them to create feature extractor by only using prompting and a feature scorer. The LLM is shown examples of best scoring features and their score and instructed to find better ones.

[1] https://www.nature.com/articles/s41586-023-06924-6

this work applies this principle to constructing decision trees, the main difference being that now we use this method to find new features at each decision point rather than global features. This brings its own set of difficulties that the paper tackles.

**Strengths:**

1. The basic premise of FunSearch is daring and interesting and this work proposes logical evolution of it that shares its properties. Its good to see more work in this direction

2. The idea of using decision tress / random forests makes a lot of sense since they are very successful for a number of problems, but are limited by the pool of features the can pick from

**Weaknesses:**

1. The use of SoTA LLM like GPT-5 in the proposed model can be a big distortion factor against other neural network baselines, since the are very performant. There should be a FunSearch baseline using GPT-5 for comparison and other weaker baselines that may leverage GPT-5 like a plain non-iterative "instruct GPT-5 to solve problem using code and the same prior knowledge as in the other (i.e. API specs)".

2. Some claims about the advantages seem inconsistent or not very convincing. The paper states neural networks "are highly data intensive" and "their ability to generalize drops drastically when in-domain data are scarce", but uses pre-trained NNs in in a way that directly contradicts this i.e. as few-shot generators of features that seem to generalize well. The method is proposed as "flexible paradigm for learning interpretable representations end-to-end" but also claims " our experiments in chess can use up to 50k lines of LLM-generated code" which seems hardly interpretable.

**Questions:**

How would FunSearch with GPT-5 perform here?

---

> ### Author Response · Authors · 2025-11-21
>
> We thank the reviewer for the constructive feedback! We answer both concerns below.
>
> > Some claims about the advantages seem inconsistent or not very convincing. The paper [...] claims " our experiments in chess can use up to 50k lines of LLM-generated code" which seems hardly interpretable.
>
>
> Thank you for pointing this out! We rephrased both claims to be more specific (see Section 1, edits in red). In particular, neural networks are data intensive when trained as X -> Y predictors for the task at hand (e.g., agreeing with our results with Transformers in Chess, or fine-tuned language model heads in Text Classification, which are both baselines that LeaPR outperforms). We instead use LLMs for a task that they have been extensively trained on (Python code generation), and the universality of code as a representation enables transfer learning.
>
> Moreover, we claim that our predictors can be interpretable even if they can grow very large, because they are modular compositions of small building blocks (the features) that we have found to be generally easy to understand. This is in the same way that large software (such as your browser, with millions of lines of code) is human-interpretable – that does not mean that it can be understood immediately by anyone (which would imply that anything interpretable has to be extremely simple), but rather that it enables human inspection of how it accomplishes the task. Our case study using SHAP values (Section 4.5) shows how simple interpretability tools allow one to make sense of LeaPR models – both individual predictions and general patterns the model learned to rely on — without needing to read all the features at once. We made this point more clear when we mention the size our models can have. Thank you for pointing this out and helping make the argument clearer!
>
> > How would FunSearch with GPT-5 perform here?
>
> This is indeed a great suggestion. We have both ran FunSearch and a Program of Thoughts (PoT) [1] baseline for all domains, both with GPT-5.1 (released N days ago). In PoT, we sample K proposals of predictors, written as code, from the LLM, choose the predictor with the best training error, and evaluate it on the test set. FunSearch starts similarly, but we adapted the baseline evolutionary procedure from Google's public implementation to iteratively evolve the predictors with "islands". These both serve as baselines that can leverage the prior knowledge of a much stronger model than the ones we used, without the added learning structure provided by LeaPR methods. The new results (see updated tables in Section 4) show that even FunSearch with GPT-5.1 underperforms across all domains, despite using a model with much stronger prior knowledge than gpt-5-mini (except for chess, FunSearch with GPT-5.1 also underperforms LeaPR with gpt-4o-mini).
>
> This result emphasizes our claim that learning in the LeaPR hypothesis class scales much better than trying to evolve a single monolithic predictor. To illustrate this point, here is a snippet of the highest-scoring predictor that GPT-5.1 evolves in Ghostbuster (it has 300 lines in total):
>
> ```python
> def predict(text: str) -> bool:
>     import re
>     t = text.strip()
>     lower = t.lower()
>     # Heuristic 0: Very short snippets – often AI completions or prompts
>     if len(t) < 40:
>         return 1.0
>     score = 0.0
>     # ...
>     # Heuristic 2: Repetition of formulaic phrases
>     repetitive_phrases = ["in conclusion", "overall", "this essay", ] # ... many more
>     rep_hits = 0
>     for p in repetitive_phrases:
>         if lower.count(p) > 1: rep_hits += 1
>     if rep_hits >= 1: score += 1.0
>     if rep_hits >= 2: score += 1.0
>     #...
>     # New Heuristic 20: Very balanced paragraph lengths in mid-long essays (AI-ish)
>     if 500 < len(t) < 2000 and len(paragraphs) >= 4:
>         p_lengths = [len(p.split()) for p in paragraphs]
>         avg_p = sum(p_lengths) / len(p_lengths)
>         near_p = sum(1 for L in p_lengths if abs(L - avg_p) <= 25)
>         if near_p / len(paragraphs) > 0.8:
>             score += 0.5
>     # Final decision threshold (tuned to reduce over-prediction of AI)
>     return 1.0 if score >= 3.6 else 0.0
> ```
>
> It is easy to see why directly optimizing complex predictors like this will hardly scale: GPT-5.1 had to jointly (a) extract useful features out of the input text, and (b) decide how to combine these features into a final prediction. Even if the features are sensible, the space of possible ways to combine them is just too large for evolutionary search to be an effective optimizer. In LeaPR, we drastically simplify the model's task by eliminating step (b) from what the LLM has to accomplish, and replacing that step with random forest training – since with good features it’s easy enough to obtain a good predictor in this way.
>
> We hope this result and example help address your concerns. Please let us know if there's anything else we can provide!

---

### Meta-Review · Area_Chair_zmQR · 2026-01-07

**Summary:**

While the authors were able to address most clarity questions and the reviewers found the ideas interesting, in the end the paper still had three major concerns:

1. The paper lacks several reasonable and low-effort LLM baselines
2. The claims of interpretability that are not rigorously supported with experimentation.
3. The experimental settings are far too simple to be viewed as practical (e.g. MNIST-era).

Secondarily, reviewers tended to want to see some more exploration of the new model class.

For these reasons I recommend rejection.

**Reviewer Concerns:**

Concerns addressed:
+ A reviewer found the paper to make conflicting statements about data (scaling to big data but also being data efficient) which the authors clarified.
+ Key details surrounding the algorithm were initially missing (i.e. definitions of subroutines), which were added during the discussion phase.
+ Insufficent baselines. A common criticism is that the proposed work does not compare to obvious baselines using the same LLMs in more standard ways, either via various prompting paradigms or via LLM powered Funsearch naive baselines. This was addressed by adding these baselines as experiments in the response.
+ A reviewer felt the method had limited novelty as other works have used LLMs to generate features. The authors clarified by pointing out that their method uses LLMs to generate code that generates features, which is slightly different.
+ Ablations of the parameters: A reviewer wanted to know how the various properties of leapr models (i.e. number, complexity or quality of features) affects the model performance. Since this is a new class of models, this is a reasonable thing to ask. The authors followed up with new experiments analyzing the number of features.
+ Several reviewers had concern on computational cost for generating the features, which was addressed by adding new analysis to the main paper and pointing reviewers to related sections in the appendix.

Concerns outstanding:
+ A reviewer doubted the claims of interpretability (claiming interpretable representations but then using tens of thousands of lines of generated code). The authors argued that this is still interpretable because the programs are modular and can be inspected, which doesn’t really address the main issue with 50k lines of code being hard to interpret (this is of the same flavor as blanket asserting decision trees to be interpretable even when the tree has thousands of nodes, just because one can technically inspect the decision tree). In fact, the authors response further reinforces this problem as they have to actually use SHAP (an external, posthoc interpretability tool) to reinterpret their leapr models, which supports that the proposed leapr models are not as obviously interpretable as claimed.
+ Remaining baselines: While two baselines were added, obvious baselines of direct LLM prompting (with or without chain of thought) were still ommitted, even though explicitly suggested by the reviewers and is technically easier to run than the baselines that the authors ran. I am ignoring the suggestion to run TabPFN as the authors correctly point out that it is not applicable. However, there are still simple LLM baselines that likely should have been run.
+ The ablation on number of features does not address the concern about how the quality of features affects performance. That complex features tend to help more is stated without supporting evidence.
+ Unclear practical use: the chosen tasks to evaluate on, while benchmark tasks, were perhaps a little too simple for most reviewers to believe that this would directly translate to complex tasks (e.g. all image experimentaiton is on MNIST variants). The authors argue that one task (detection of AI generated text) is highly practical, but is still a simple task given that the neural baselines have already effectively solved this problem and leapr cannot do better. For another reviewer, they added results on SNLI to up the complexity, but ended up with subpar results that are worse than simple heuristics.
+ The choice of using decision trees is arbitrary, and the authors acknowledge this as future exploration.
+ While the relation to feature learning work was clarified in the response/discussion, reviewers felt teh paper itself would benefit from addressing this directly.

**Reviewer Scores:**

Reviewer idVN likely would have increased their score slightly (from 4 to 6) as the baselines they asked for were the ones that were actually asked for, and their other concern was addressed.

Reviewer KrKR likely would have increased their score slightly (from 2 to 4) as some of their concerns, but not all, were addressed. As some unaddressed concerns related to baselines, which is typically a major concern, this would have likely stayed negative.

Reviewer iR4d likely would have increased their score slightly (from 4 to 6) as they had a number of concerns addressed and was most amenable to being open to new methodologies that don’t necessarily perform well, however would have still had reservations regarding ablations and the relation to other work.

Reviewer kY8q likely would not have changed their assessment (2) as their concerns are supported by other reviewers even if the specifics of their criticisms were not quite relevant to the submission.

---

### Decision · Program_Chairs · 2026-01-26

Reject